# Retrotransposon LINE-1 bodies in the cytoplasm of piRNA-deficient mouse spermatocytes: Ribonucleoproteins overcoming the integrated stress response

Chiara De Luca[1], Anuj Gupta[2], Alex Bortvin[1]*

1 Department of Embryology, Carnegie Institution for Science, Baltimore, Maryland, United States of America, 2 The Sidney Kimmel Comprehensive Cancer Center, Johns Hopkins University School of Medicine, Baltimore, Maryland, United States of America

* bortvin@carnegiescience.edu

## Abstract

Transposable elements (TE) are mobile DNA sequences whose excessive proliferation endangers the host. Although animals have evolved robust TE-targeting defenses, including Piwi-interacting (pi)RNAs, retrotransposon LINE-1 (L1) still thrives in humans and mice. To gain insights into L1 endurance, we characterized L1 Bodies (LBs) and ORF1p complexes in germ cells of piRNA-deficient *Maelstrom* null mice. We report that ORF1p interacts with TE RNAs, genic mRNAs, and stress granule proteins, consistent with earlier studies. We also show that ORF1p associates with the CCR4-NOT deadenylation complex and PRKRA, a Protein Kinase R factor. Despite ORF1p interactions with these negative regulators of RNA expression, the stability and translation of LB-localized mRNAs remain unchanged. To scrutinize these findings, we studied the effects of PRKRA on L1 in cultured cells and showed that it elevates ORF1p levels and L1 retrotransposition. These results suggest that ORF1p-driven condensates promote L1 propagation, without affecting the metabolism of endogenous RNAs.

## Author summary

Transposable elements (TEs) are mobile DNA sequences whose proliferation endangers the integrity of the host cell and its genome. Although cells have acquired sophisticated TE-targeting defenses, TEs continue to thrive in all life forms. To address this fascinating question, this study focused on a particular type of TE known as retrotransposon LINE-1. This work shows that LINE-1-expressing germ cells of mice develop cytoplasmic aggregates (LINE-1 bodies or LBs) enriched in LINE-1, RNAs, and proteins implicated in the negative regulation of RNA expression. Despite the overwhelming repressive nature of LBs, the integrity and translation of LB-localized RNAs are not affected. RNA expression appears to stay intact because LINE-1 has adapted to interact with the PRKRA protein that relays stress signals to downstream proteins, triggering general translational

**Data Availability Statement:** The sequencing data discussed in this publication have been deposited in NCBI's Gene Expression Omnibus and are accessible through GEO Series accession numbers

 

GSE222415 (https://www.ncbi.nlm.nih.gov/geo/query/acc.cgi?acc=GSE222415) and GSE222416 (https://www.ncbi.nlm.nih.gov/geo/query/acc.cgi?acc=GSE222416). • The mass spectrometry proteomics data have been deposited to the ProteomeXchange Consortium via the PRIDE partner repository with the dataset identifier PXD039425. • Original code is available at Carnegie Embryology GitHub (https://github.com/ciwemb/bortvin-2023-orf1p).

**Funding:** This work was supported by Carnegie Institution for Science Endowment (AB) and National Institutes of Health grant R21HG010512 (AB). The funders had no role in study design, data collection and analysis, decision to publish, or preparation of the manuscript. Both AB and CDL received salaries from Carnegie Institution for Science with some funds coming from the NIH award.

**Competing interests:** The authors have declared that no competing interests exist.

inhibition. Indeed, the simultaneous expression of LINE-1 and PRKRA in cultured cells elevates LINE-1 protein expression and mobilization. Our data suggest that LINE-1 has evolved a mechanism to counteract the translational shut-off, thus increasing its chances of propagation without compromising the host cell.

## Introduction

Transposable elements (TEs) are discrete DNA sequences that can spread throughout the host genome [1,2]. TE mobilization fuels the evolution of genomes and reprograms gene expression regulatory networks [3–5]. However, TE insertions also mutate genes, perturb gene expression and disrupt genome integrity [6–8]. Previous studies have revealed the existence of defensive mechanisms aimed at reducing the destructive potential of TEs. Among these are DNA methylation [9,10], repressive histone modifications [11], small Piwi-interacting (pi)RNAs [12,13] and repressive Kruppel-associated box-containing zinc-finger proteins (KRAB-ZFP) [14–17]. Despite the long evolutionary history of TE-targeting mechanisms [9,18], most eukaryotic genomes, including human and rodent genomes, still harbor active TEs [19–22]. How TEs resist and adapt to constant pressure from hosts is a fascinating question deserving further analysis [23]. Prior studies revealed a few examples of arms races between TEs and their hosts, including mutations and deletions of KRAB-ZFP binding sites from regulatory sequences of target TEs [15,23,24]. Unlike viruses, however, TEs rarely encode active countermeasures allowing to escape the defensive mechanisms of the host [23]. In particular, no such mechanisms have been reported for TEs active in animal genomes.

To better understand the interactions of TEs with their hosts in mammals, we focus on retrotransposon **L**ong **In**terspersed **E**lement-1 (LINE-1 or L1) [25]. L1 sequences are highly abundant in mammalian genomes accounting for 17.4% of human and 19% of mouse genomes [26,27]. Crucially, both genomes possess intact L1 copies capable of retrotransposition [20–22]. Indeed, an ongoing L1 activity is evident in the human population, with 124 L1-mediated disease-causing mutations identified so far [7]. The L1 life cycle involves the transcription and translation of L1 RNA, producing the RNA-binding protein ORF1p and the endonuclease/reverse transcriptase ORF2p [8,25,28]. The two proteins bind L1 RNA forming a ribonucleoprotein particle (RNP) [29,30]. Once L1 RNP enters the nucleus, ORF2p uses its two activities to execute target-primed reverse transcription (TPRT), generating a new L1 copy [31,32].

Much of the current understanding of L1 comes from studies in cultured mammalian cells ectopically overexpressing L1. This approach bypasses L1 repression, allowing mechanistic studies and quantification of L1 retrotransposition [33–35]. Furthermore, epitope tagging of L1 proteins allows the visualization of their subcellular distribution and the characterization of isolated L1 RNPs [36–40]. For example, prior studies documented the co-localization of ORF1p with protein markers of stress granules (SGs), RNA-containing biocondensates that appear upon cell exposure to various stresses [41–43].

The discovery of a germ cell defensive piRNA pathway extended systematic TE studies to various animal species. This adaptive immunity protects germ cells and their genomes from TEs [12,13]. Critically, because piRNAs function predominantly in germ cells, genetic perturbations of the piRNA pathway render mutant animals sterile but viable. One conserved piRNA factor is *Maelstrom* (*Mael*), which encodes an RNA-binding protein required for piRNA biogenesis [44–48]. Male germ cells of *Mael*$^{-/-}$ mice express high levels of L1, making them an ideal model to characterize the consequences of L1 overexpression [45]. In our prior studies,

we characterized *Mael* expression and examined the impact of L1 overexpression on germ cells [45,49,50]. We also used L1 ORF1p overexpression in the *Mael* mutant to discover novel features of germ cell differentiation in both sexes [51–53]. This work focused on cytoplasmic aggregates highly enriched in ORF1p that form in *Mael^-/-* male germ cells. Using a combination of approaches, we investigated the macromolecular composition of ORF1p complexes and revealed a mechanism that allows L1 to evade post-transcriptional repression.

## Results

### *Mael^-/-* spermatocytes develop cytoplasmic bodies enriched in ORF1p, L1 RNA, and ribosomes

Meiotic male germ cells (spermatocytes) of *Mael^-/-* mice overexpress L1 and develop prominent ORF1p-positive cytoplasmic bodies packed with ~25 nm electron-dense particles [45]. This work will refer to these structures as LBs for **L**INE-1 **B**odies (a term also suggested recently by others [54]). In EM images, LBs first appear as submicron cytoplasmic aggregates (Fig 1A, 1B, 1A' and 1B'), reminiscent of ORF1p aggregates occurring in wild-type (WT) spermatocytes (S1 Fig). In *Mael^-/-* mice, smaller LBs appear to fuse, forming large LBs that sometimes associate with membranes (Fig 1C–1E). LBs reach 3 μm in size, exceeding the maximum sizes reported for Stress Granules (SGs, 2 μm) and Processing Bodies (p-bodies or PBs, 0.5 μm) [55,56]. A 3D reconstruction of LBs, using multiple z-plane confocal images of ORF1p signal in *Mael^-/-* spermatocytes, revealed that LBs occupy large cytoplasmic volumes while embracing the nuclei (Fig 1F).

We next asked if electron-dense particles comprising LBs corresponded to L1 RNPs [57], ribosomes, or both. Previously, we used immunogold-EM to confirm the predominant ORF1p localization to LBs [45]. To verify L1 RNA enrichment in LBs, we performed sequential L1 ORF1p immunofluorescence followed by Hybridization Chain Reaction (HCR) RNA FISH using L1 DNA probes [58]. We observed robust L1 RNA signals that colocalized with L1 ORF1p in LBs of *Mael^-/-* spermatocytes (Fig 1G). Control spermatocytes sporadically have cytoplasmic L1 RNA and ORF1p-positive granules similar to early LBs (Figs 1A', 1G, and S1).

To assess the ribosomal content of LBs, we examined the subcellular localization of RPS6 and RPL28 proteins of small and large ribosomal subunits, respectively. Immunofluorescence analysis revealed the abundant and uniform distribution of the two tested ribosomal proteins throughout individual LBs (Fig 1H). The above findings suggest that LBs are large cytoplasmic compartments enriched in L1 RNA, ORF1p, and ribosomes. Although LBs might be cytoplasmic RNA granules, they are distinct from SGs and PBs that lack ribosomes [59].

### L1-polysome and L1 RNP complexes in *Mael^-/-* testes

To investigate L1 RNA and ORF1p association with ribosomes, we fractionated *Mael^-/-* and *Mael^+/-* testicular extracts through 10–50% linear sucrose gradients. We collected twelve fractions per gradient from the top (fraction 1) to the bottom (fraction 12) and analyzed the distributions of proteins and RNAs by Western blot and quantitative RT-PCR, respectively. In one set of experiments, we added cycloheximide (CHX) to samples to prevent ribosome runoff. We validated the resulting gradients by monitoring the 254 nm UV absorbance profile to reveal the characteristic peaks corresponding to 40S and 60S ribosomal subunits, 80S ribosomes, and polysomes (Fig 2A). Western blot analysis of ribosomal protein RPS6 confirmed the ribosome distribution in the gradient. At the same time, all gradient fractions contained ORF1p but with a clear enrichment in polysome-containing fractions toward the bottom of the gradient (Fig 2A). Importantly, qRT-PCR analysis of L1 RNA using *ORF1* amplicon

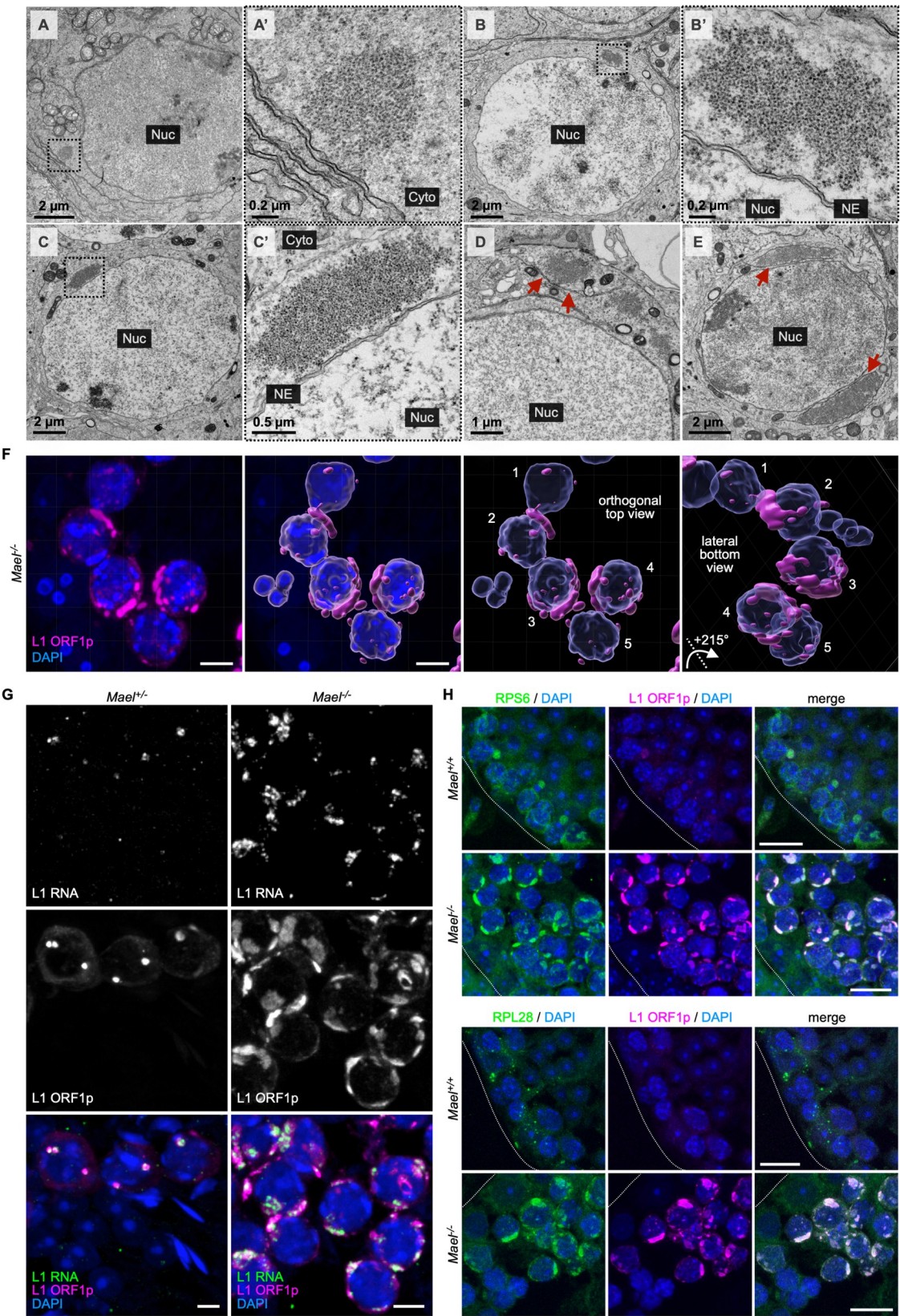

**Fig 1. L1 bodies (LBs) are large cytoplasmic compartments enriched in ORF1p, L1 mRNA, and ribosomes. (A–E)** Electron micrographs exemplifying the formation of LBs in *Mael*$^{-/-}$ spermatocytes. Boxed areas in A and B identify small cytoplasmic aggregates magnified in A' and B', respectively. Boxed area in C identifies an intermediate size LB magnified in C'. D and E show more prominent LBs, partially surrounded by a bilayer membrane (red arrows). Nuc: nucleus. Cyto: cytoplasm. NE: nuclear envelope. **(F)** Immunofluorescence staining of L1 ORF1p on *Mael*$^{-/-}$ spermatocytes shows ORF1p accumulation in LBs (left panel, magenta). 3D reconstruction of LBs (magenta) and nuclei (blue) (middle and right panels); numbers [1–5] are used to label corresponding cells. Scale bars: 5 μm. **(G)** HCR RNA-FISH of L1 RNA (green) and immunofluorescence staining of L1 ORF1p (magenta) on *Mael*$^{+/-}$ and *Mael*$^{-/-}$ testis sections. Scale bars: 5 μm. **(H)** Double immunofluorescence staining of ribosomal protein S6 (RPS6, top two rows, green) or L28 (RPL28, bottom two rows, green) and L1 ORF1p (magenta) showing their colocalization in LBs. Dotted white lines mark seminiferous tubule boundaries. Scale bars: 15 μm. See also S1 Fig.

showed a corresponding broad distribution pattern but with a peak in early polysomal fractions 6–8 (Fig 2B). The control beta-actin (*Actb*) mRNA's sedimentation profile confirmed the efficiency of the CHX treatment. Consistent with previous studies [57,60,61], the observed L1 RNA gradient pattern suggests the existence of L1 ribonucleoprotein (RNP) complexes distinct from ribosome-L1 RNA associations.

To assess how the association with ribosomes influences ORF1p and L1 RNA sedimentation, we fractionated *Mael*$^{-/-}$ testicular extracts in the presence of 10 mM EDTA, causing ribosome dissociation. Indeed, both the UV absorbance profile (Fig 2C, in blue) and the RPS6 sedimentation behavior showed robust changes following EDTA treatment of the sample (Fig 2C, + EDTA). In contrast, ORF1p showed only modest changes in its distribution across the gradient (Fig 2C, + EDTA) compared to the previously described cycloheximide treatment. Furthermore, while *Actb* mRNA exhibited the expected shift from polysomes to lighter fractions, L1 RNA showed enrichment in the same region where ORF1p sediments (Fig 2D, fractions 5–8). These results further suggest the existence of a population of L1 RNA- and ORF1p-containing complexes independent from ribosomes. Finally, dual RNase A/T1 digestion of EDTA-treated samples disrupted such L1 complexes (Fig 2C, + EDTA + RNase), confirming the central role of RNA in generating the ORF1p pattern in presumptive RNP regions of the gradient. These data suggest that the cellular pool of L1 RNA and ORF1p comprises ribosome-independent and ribosome-associated complexes. The former might correspond to assembled L1 RNPs, while the latter could be the translating L1 polysomes.

## L1 ORF1p associates with L1, other TE and genic mRNAs

To characterize ORF1p complexes, we first identified the RNA present in EDTA-resistant RNPs. We performed anti-ORF1p immunoprecipitation from pooled sucrose fractions 5–8, followed by RNA-seq of the precipitated material (Figs 2C and S2). Besides L1 RNA, non-autonomous retrotransposon RNAs and genic mRNAs can form RNPs with L1 proteins to re-enter the genome [36,62,63]. We therefore examined our RNA-seq data initially for genomic repeat RNAs and subsequently for genic mRNAs. We compared the RNA content of ORF1p immunoprecipitated (IP) samples to that of the unfractionated *Mael*$^{-/-}$ testis lysates (TOTAL), pooled fractions 5–8 (INPUT), or material bound non-specifically to carrier beads only (BO). We observed a strong enrichment of RNAs corresponding to evolutionarily young L1MdA and L1MdT families, which account for most full-length, intact L1 copies in the mouse genome (Fig 3A). Despite comparable levels in the TOTAL and INPUT samples, RNA of an evolutionarily older L1MdF2 family showed only minor enrichment in IPs. This observation further supports the notion that ORF1p and L1 RNA in fractions 5–8 form RNPs with a strong *cis*-preference for the coding L1 RNA, as reported previously [29,30].

RNA-seq analysis also revealed two unexpected features of germ-cell ORF1p complexes. First, in contrast to the established role of L1 in the mobilization of non-autonomous retrotransposons [64,65], we observed no enrichment for RNA of SINE B1 and B2 elements, despite

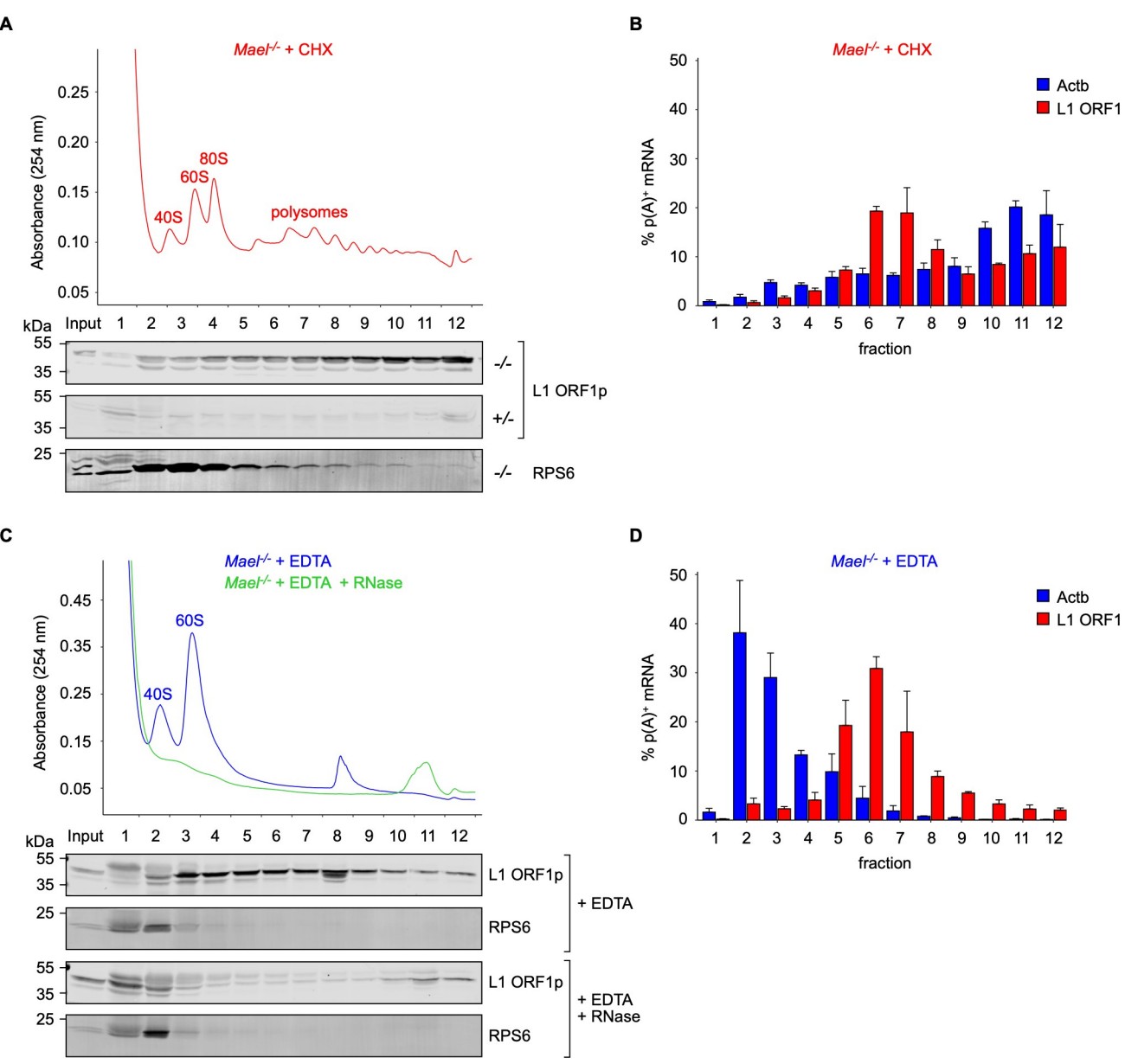

**Fig 2. L1-polysome and L1 RNP complexes in *Mael*^-/- testes. (A)** Western blot analysis of L1 ORF1p and RPS6 in *Mael*^-/- testicular extracts fractionated through 10–50% sucrose gradients in the presence of cycloheximide (CHX). Fractionated *Mael*^+/- extract served as a negative control. The absorbance profile of the gradient is shown in red. **(B)** qRT-PCR analysis of L1 RNA (L1 *ORF1* amplicon) and beta-actin mRNA (*Actb*) in *Mael*^-/- testicular extracts fractionated through 10–50% sucrose gradients in the presence of CHX. RNA amounts per fraction are reported as percentages of the total RNA detected across the gradient; barplot shows mean percentage values + SEM of four biological replicates. **(C)** Western blot analysis of L1 ORF1p and RPS6 in *Mael*^-/- testicular extracts fractionated through 10–50% sucrose gradients in the presence of EDTA (+ EDTA); the absorbance profile of the gradient is shown in blue. Upon addition of RNases A and T1 (+ EDTA + RNase) ribosomes are degraded (profile in green) and L1 RNPs disaggregated. **(D)** qRT-PCR analysis of L1 RNA (L1 *ORF1* amplicon) and beta-actin mRNA (*Actb*) in *Mael*^-/- testicular extracts fractionated through 10–50% sucrose gradients in the presence of EDTA. Barplot shows mean percentage values + SEM of three biological replicates. See also S2 Fig.

their high abundance in the TOTAL (Fig 3A). Second, endogenous retrovirus MMERVK10C-int RNA showed strong enrichment in ORF1p complexes (Fig 3A). Nevertheless, numerous other TE RNAs, including those of endogenous retrovirus IAP, did not enrich in ORF1p immunoprecipitates (Figs 3A and S3A).

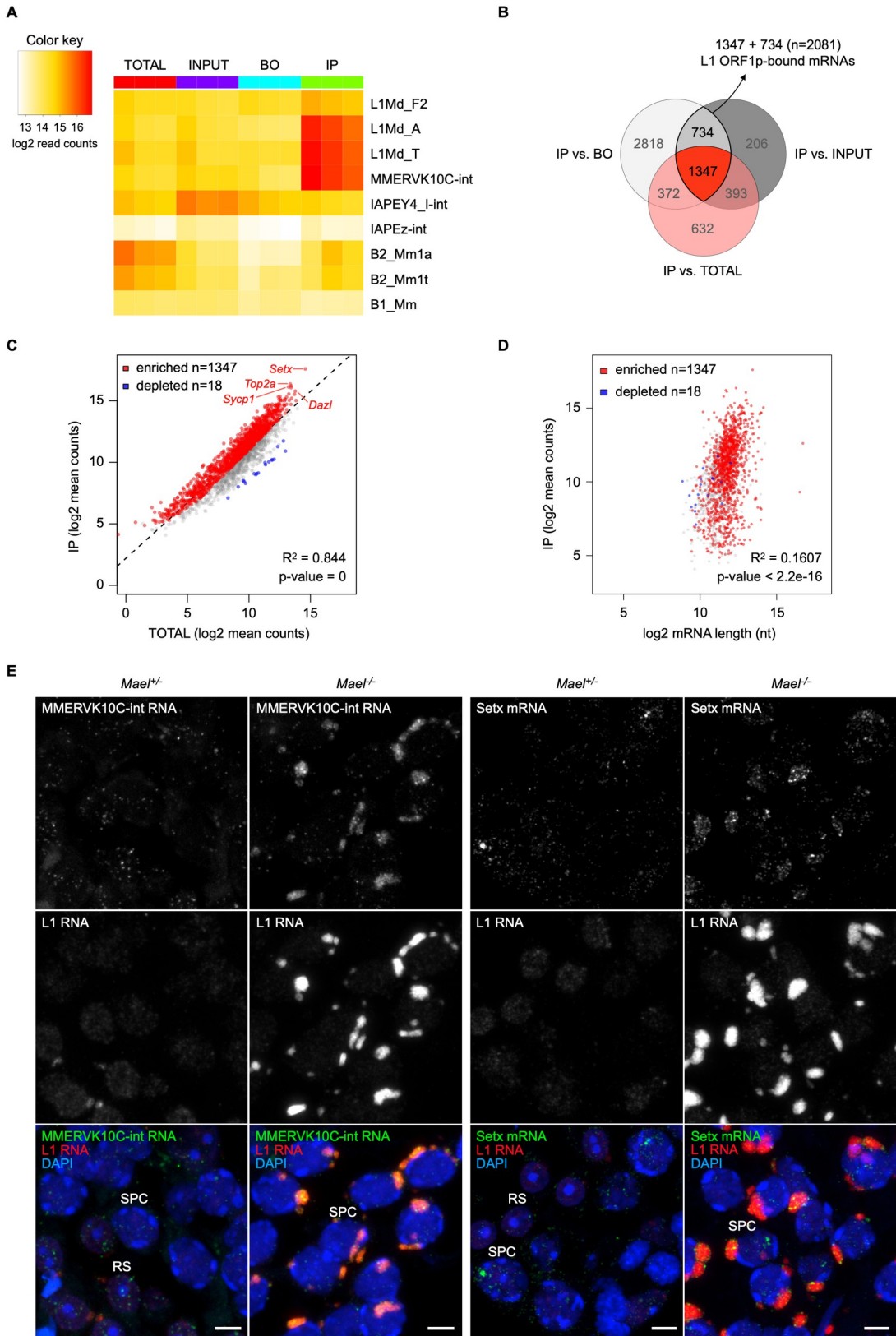

**Fig 3. L1 ORF1p binds to L1, other TEs and abundant genic mRNAs. (A)** RNA levels of selected genomic repeats across anti-ORF1p co-immunoprecipitation samples (TOTAL, INPUT, Beads Only (BO), IP; in triplicate). Heatmap reports log2

normalized collapsed read counts; color indicates RNA levels. **(B)** Identification of mRNAs bound by L1 ORF1p in IP samples (n = 2081). For every comparison (IP vs. BO, IP vs. TOTAL or IP vs. INPUT), mRNAs with log2FoldChange > 1 and padj < 0.05 were considered enriched. **(C)** Linear regression analysis of n = 2081 ORF1p-bound mRNAs showing the relationship between their expression level in *Mael*$^{-/-}$ testes (x axis, TOTAL) and their enrichment in L1 RNPs (y axis, IP). Scatterplot shows log2 mean normalized counts. **(D)** Linear regression analysis of n = 2081 ORF1p-bound mRNAs showing the relationship between their length (x axis, log2 length in nucleotides) and their enrichment in L1 RNPs (y axis, IP). **(E)** Multiplexed HCR RNA-FISH of MMERVK10c-int and Setx with L1 RNA on *Mael*$^{+/-}$ and *Mael*$^{-/-}$ testis sections. Images of spermatocytes (SPC) and round spermatids (RS) are reported. Scale bars: 5 μm. See also S3 and S4 Figs.

Focusing on genic mRNAs, we identified 2081 transcripts associated with ORF1p (log2FoldChange > 1 and padj < 0.05 in IP vs. BO & IP vs. INPUT) (Fig 3B and S1 Table). Of these, 1347 mRNAs were enriched over the TOTAL, consistent with their ORF1p association (Fig 3B and 3C, in red). The remaining 734 mRNAs (shown in grey and blue in Fig 3C) did not show enrichment over the TOTAL. Among them, a small group of "depleted" mRNAs (n = 18; Fig 3C, in blue) showed higher levels in *Mael*$^{-/-}$ testis extracts (x-axis, TOTAL) than in ORF1p immunoprecipitates (y-axis, IP).

To gain insight into mRNA-ORF1p associations, we examined the relationships between the abundance of mRNAs in ORF1p complexes and their expression levels and lengths. mRNA expression levels correlated strongly with the probability of associating with ORF1p ($R^2$ = 0.844, $p$ = 0; Fig 3C). Prior studies showed that mRNAs enriched in cytoplasmic granules are longer than an average transcript [43,66]. However, no such correlation is apparent for ORF1p-associated mRNAs (Fig 3D), further highlighting the difference between LBs versus SGs and PBs. The simplest explanation of these results is that an individual mRNA expression level is the primary determinant of their association with ORF1p. We also tested if enriched mRNAs (n = 1347) belonged to a functionally focused subset of the germ cell transcriptome. Although Gene Ontology analysis revealed the expected over-representation of germ cell and cell cycle mRNAs, no functional categories appeared to dominate (S3B Fig and S2 Table).

To verify the localization of ORF1p-associated RNAs in LBs, we performed an HCR RNA FISH of *MMERVK10c-int* and *Setx* mRNAs. We observed the colocalization of both mRNAs with L1 RNA in LBs (Fig 3E). In control *Mael*$^{+/-}$ samples, both mRNA species showed a diffuse dotted pattern. Additional negative (RNase A treatment before hybridization) and positive (U6 small nuclear RNA hybridization) controls for the RNA FISH experiments confirmed the specificity of the detection (S4 Fig). These results strongly support our RNA-seq findings and confirm the capability of ORF1p to bind to RNAs independently of their functionality or length and amass them in LBs.

## ORF1p interacts with components of RNA granules, the CCR4-NOT complex, and ribosomes

To further characterize ORF1p interactions, we identified proteins co-immunoprecipitated with ORF1p from total *Mael*$^{-/-}$ testicular extracts using mass spectrometry. Two independent experiments detected 569 and 655 proteins, including ORF1p (S3 Table). Upon filtering out low-specificity proteins (see Materials and Methods and S3 Table), we identified 80 high-confidence ORF1p interactors common to both datasets (S4 Table). Of these, 36 are homologs of human proteins reported previously as putative ORF1p interactors (S5 Table).

To better understand the ORF1p interactome, we used the STRING database (Search Tool for Recurring Instances of Neighboring Genes, https://string-db.org/) to generate protein networks [67]. This analysis showed that ORF1p interactors common between previous studies and this work (Fig 4, "known") fell into cytoplasmic "RNA granules" and "Ribosomal proteins"

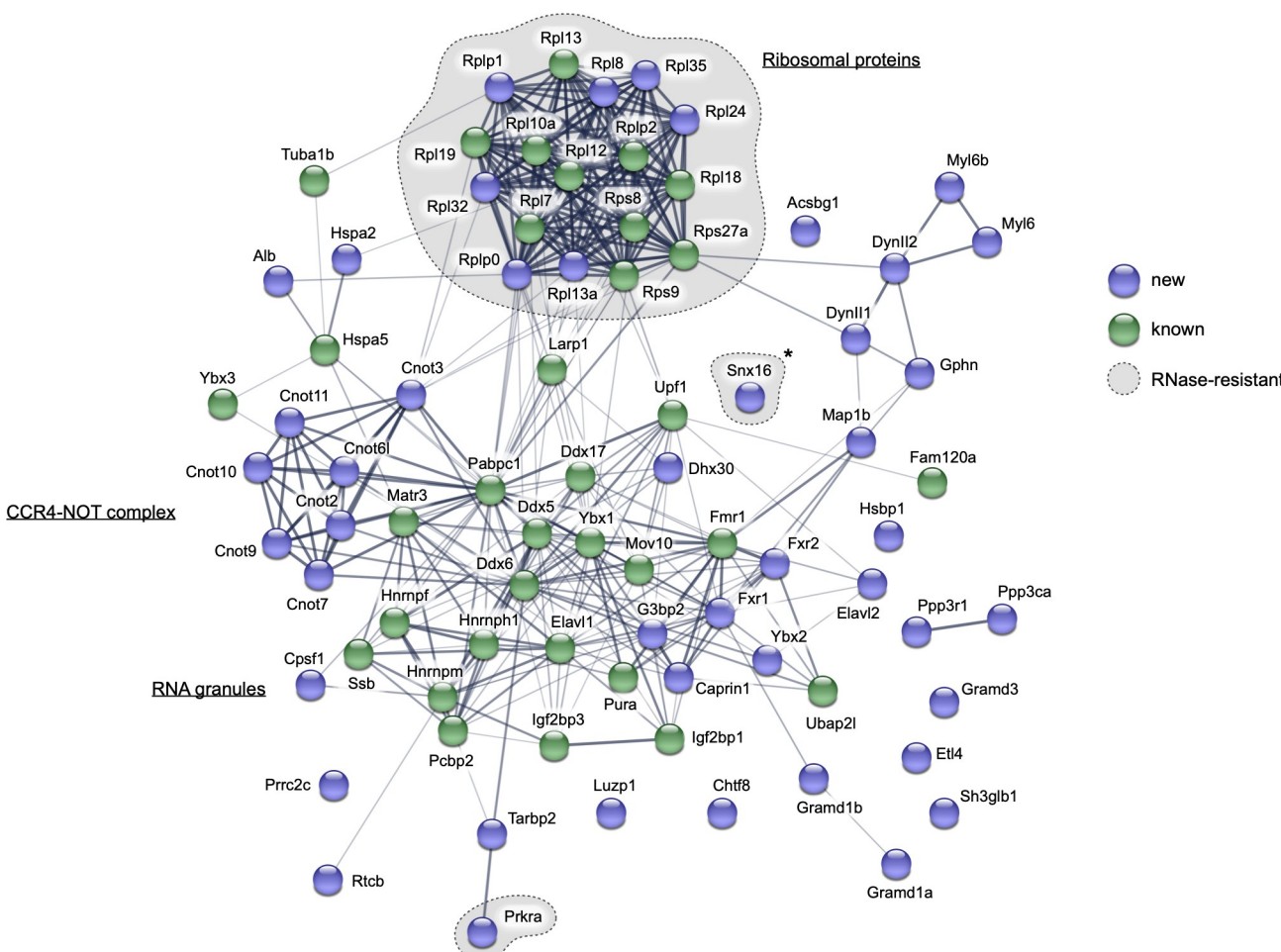

**Fig 4. L1 ORF1p associates with cytoplasmic RNA granules components, the CCR4-NOT complex and ribosomal proteins.** Network of physical and functional interactions among the 80 high-confidence L1 ORF1p interactors identified in this study. The network was generated using the STRING database; thickness of connection lines indicates the strength of data support. Proteins identified for the first time in this work (new) are marked in blue, while previously known ORF1p interactors are marked in green. ORF1p interactors resistant to RNase treatment are enclosed in dashed grey areas. *: unspecific interactor. See also S5 Fig.

networks. The "RNA granules" network includes, among others, LARP1, G3PB2, MOV10, DDX6, FMR1, YBX1, ELAV1, and PABPC1 proteins. Many of these proteins are exclusive for SGs, known to harbor ORF1p upon L1 overexpression in cultured cells [41,59]. Newly discovered ORF1p interactors associated with the same network include CAPRIN1 and FMR1-related FXR1 and FXR2 proteins, further strengthening the view of ORF1p association with SGs [68]. Newly discovered ORF1p interactors also included several proteins of the CCR4-NOT complex, a global regulator of mRNA metabolism frequently recruited to PBs [69,70]. Among the ORF1p-associated CCR4-NOT proteins, we consistently detected catalytic CNOT6L and CNOT7 proteins involved in mRNA deadenylation but not the scaffold CNOT1 protein and CNOT4 E3 ubiquitin ligase. The "Ribosomal proteins" network comprised proteins of both ribosomal subunits. Seven of eleven previously identified [36] and six newly discovered ORF1p-associated ribosomal proteins are components of the 60S subunit. This finding suggests that LBs are distinct from SGs, which contain the small (40S) but not large (60S) ribosomal subunits, and PBs, devoid of ribosomes altogether [42].

To determine the RNA-dependency of ORF1p interactions, we treated $Mael^{-/-}$ testis extracts with an RNase cocktail before immunoprecipitation (S5A and S5B Fig). The RNase treatment allowed for a better recovery of the bait protein ORF1p compared to an untreated sample (S5C Fig). This result can be explained by the removal of physical hindrance around ORF1p caused by RNA and RNA-dependent interactors. ORF1p association with most RNA granule components and the CCR4-NOT complex is strongly reduced or completely lost upon RNase treatment (S4 Table). In contrast, RNase treatment did not significantly affect the interactions of ORF1p with ribosomal proteins, despite the compromised integrity of rRNA and ribosomes (Figs 4 and S5D and S4 Table).

We next performed double immunofluorescence labeling of testicular sections to support further the identified ORF1p interactions with RNA granules proteins and the CCR4-NOT complex. We established the colocalization of RNA helicase DDX6, enriched in both PBs and SGs, and CNOT7, one of two deadenylase subunits of the CCR4-NOT complex, with ORF1p (Fig 5). DDX6 localized to cytoplasmic punctate foci in both spermatogonia and in control and ORF1p-negative $Mael$-mutant spermatocytes (Fig 5A, yellow arrows). However, DDX6 clearly redistributed to LBs whenever present in the cytoplasm of $Mael^{-/-}$ spermatocytes (Fig 5A, white arrowheads). CNOT7 shows a similar staining pattern, characterized by the formation of prominent cytoplasmic foci in the WT (Fig 5B, yellow arrows) and the aggregation with ORF1p in $Mael^{-/-}$ spermatocytes (Fig 5B, white arrowheads). We also asked if CNOT1, a scaffold protein essential for the CCR4-NOT complex assembly, localized to LBs along with ORF1p. Although absent among ORF1p interactors, we detected CNOT1 in LBs (S6A Fig, white arrowheads). Finally, given the frequent association of CCR4-NOT with PBs, we asked if LBs contained other PB proteins, such as DCP1A. Interestingly, in contrast to the abundance of DCP1A in PBs of various sizes in control samples, DCP1A appears mostly confined to prominent granules with strong signals at the periphery of $Mael^{-/-}$ seminiferous tubules (Figs 5C and S6B). Sporadically, we also observed weak diffused DCP1A staining in LBs (Fig 5C, white arrowheads). This observation suggests that L1 overexpression and LBs formation coincided with the disappearance of PBs. Cumulatively these results demonstrate that LBs are unusual hybrid cytoplasmic compartments bearing characteristics of SGs and perhaps PBs and yet enriched in ribosomes.

## ORF1p binding does not affect the stability and translation efficiency of mRNAs

The above findings raise an important question about the impact of LBs on RNA stability and translation. To test if ORF1p-associated RNAs exhibit signs of 5' or 3' degradation, we examined the read coverage across all RNAs detected in ORF1p immunoprecipitated EDTA-resistant RNPs (IP) and compared these results to the unfractionated $Mael^{-/-}$ testicular extracts (TOTAL). A reduction or lack of coverage towards the 5' or the 3' termini would be consistent with RNA degradation by resident SG and PB exonucleases. We computationally divided all detected mRNAs into ten bins, each corresponding to 10% of the total RNA length. We then calculated the read coverage across each region for every RNA and the cumulative coverage across each region for all RNAs detected in IP vs. TOTAL samples (Fig 6A). We did not observe reduction of coverage at any of the ends for immunoprecipitated RNAs compared to total extracts (Fig 6A), suggesting that ORF1p-bound RNAs are intact.

To investigate if CCR4-NOT complex association with L1 ORF1p resulted in increased deadenylation of LB-localized mRNAs, we analyzed the poly(A) tail length of two ORF1p-associated mRNAs (*Sycp1* and *Setx*) in $Mael^{-/-}$ and control juvenile testes of comparable tissue complexity at postnatal day 16 (P16) (Fig 6B and 6C). Using a PCR-based approach, we found

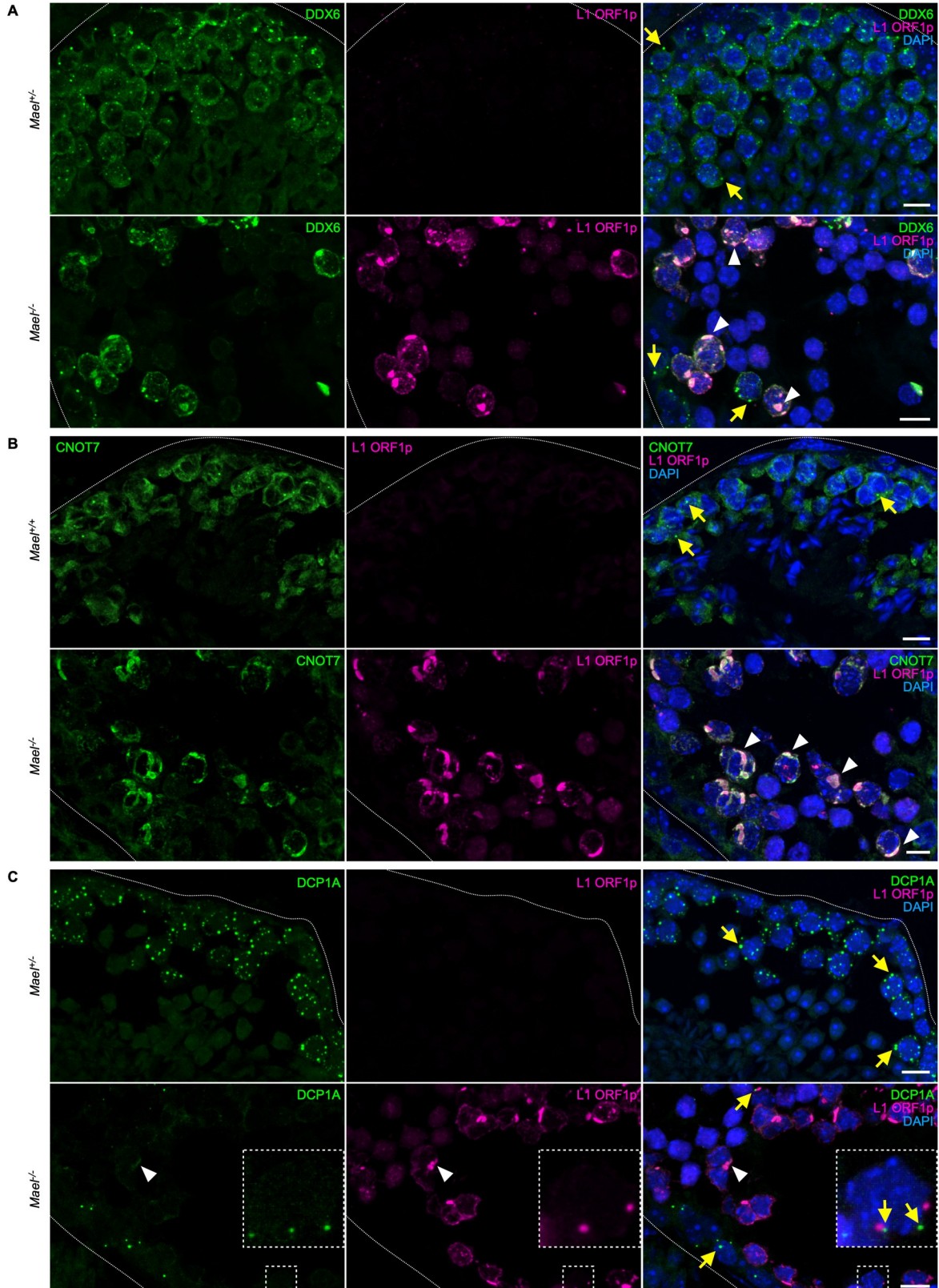

**Fig 5. LBs are enriched with DDX6 and CNOT7 proteins. (A)** Double immunofluorescence staining of RNA helicase DDX6 (green) and L1 ORF1p (magenta) in *Mael*⁺/⁻ and *Mael*⁻/⁻ testes. DDX6 is detectable in cytoplasmic foci in both *Mael*⁺/⁻ and *Mael*⁻/⁻ spermatogonia

and spermatocytes (yellow arrows); in *Mael*$^{-/-}$ both ORF1p and DDX6 concentrate in LBs (white arrowheads). Dotted white lines mark seminiferous tubule boundaries. Scale bars: 10 μm. **(B)** Double immunofluorescence staining of CCR4-NOT complex subunit CNOT7 (green) and L1 ORF1p (magenta) in *Mael*$^{+/+}$ and *Mael*$^{-/-}$ testes. CNOT7 is detectable in cytoplasmic foci in *Mael*$^{+/+}$ spermatocytes (yellow arrows); in *Mael*$^{-/-}$ CNOT7 colocalizes with ORF1p in LBs (white arrowheads). Scale bars: 10 μm. **(C)** Double immunofluorescence staining of DCP1A (green) and L1 ORF1p (magenta) in *Mael*$^{+/-}$ and *Mael*$^{-/-}$ testes. DCP1A shows a characteristic granular pattern in both samples (yellow arrows), although PBs appear overall less abundant in *Mael*$^{-/-}$ seminiferous tubules; weak diffuse DCP1A signal is also detected in LBs (white arrowheads). Boxed areas in *Mael*$^{-/-}$ panels are magnified in insets. Scale bars: 10 μm. See also S6 Fig.

no significant differences in the poly(A) tail length of the tested mRNAs between *Mael*$^{-/-}$ and control littermates (Fig 6C). These findings suggest that the CCR4-NOT complex does not deadenylate LB-localized mRNAs.

To assess the impact of LB-localized RNA granule proteins on the translation of ORF1p-bound mRNAs, we performed ribosome profiling of *Mael*$^{-/-}$ and control P16 testes. As expected, this analysis showed low expression and ribosomal coverage of the mutated *Mael* mRNA in the *Mael*$^{-/-}$ sample (Fig 6D). In contrast, we found no evidence of significant disruption of translation efficiency in general and including ORF1p-associated mRNAs (ORF1p-bound, n = 1313) (Fig 6D).

To further investigate the translational status of P16 testes, we tested the eIF2α phosphorylation level in *Mael*$^{+/-}$ and *Mael*$^{-/-}$ littermates (Fig 6E). Phosphorylation of translation initiation factor eIF2α ultimately causes inhibition of translation. In agreement with our ribo-footprint data, we found no significant difference in phospho-eIF2α levels between *Mael*$^{-/-}$ and *Mael*$^{+/-}$ testes (Fig 6E), consistently with an unperturbed translation. Collectively these data show that the ability of ORF1p to amass endogenous mRNAs to LBs does not affect their stability or translation efficiency.

## L1 coopts PRKRA to enhance ORF1p levels and retrotransposition

The above data suggest that despite numerous negative regulators of RNA expression in LBs, ORF1p-associated mRNAs avoid translational repression, suggesting a counteracting mechanism(s). Of particular interest is an RNA-independent ORF1p interactor PRKRA (also known as RAX and PACT), a protein that activates Protein Kinase R (PKR) [71]. Once activated, PKR phosphorylates eIF2α, causing translation inhibition [72]. PRKRA can also repress PKR and prevent activation of eIF2α, thus enhancing gene expression at the translational level [73,74]. In light of these opposing PRKRA activities, we asked which PRKRA-dependent mechanism operates in L1-overexpressing spermatocytes.

We first validated our earlier mass spectrometry findings by detecting PRKRA in the ORF1p immunoprecipitates from untreated and RNase-treated *Mael*$^{-/-}$ testicular lysates (Fig 7A). We also showed PRKRA and ORF1p colocalization in LBs (Figs 7B and S7A). Next, to test for PRKRA involvement in the L1 lifecycle, we generated a plasmid with an EF-1α promoter driving the expression of an mRNA encoding a 3XFLAG N-terminally tagged mouse PRKRA followed by P2A peptide and EGFP (pPRKRA; S7B Fig). We used a plasmid bearing the EGFP reporter as a negative control (pEGFP). To test if ectopic PRKRA interferes with endogenous L1 expression, we transfected 3XFLAG-tagged PRKRA into F9 cells expressing high levels of endogenous L1 elements. Ectopic PRKRA protein was detected by both anti-FLAG and anti-PRKRA antibodies with overlapping signals at the expected molecular weight; however, endogenous ORF1p expression remained unperturbed (S7C Fig). Finally, endogenous ORF1p interacted and co-localized with the PRKRA protein expressed ectopically in F9 cells (S7D and S7E Fig).

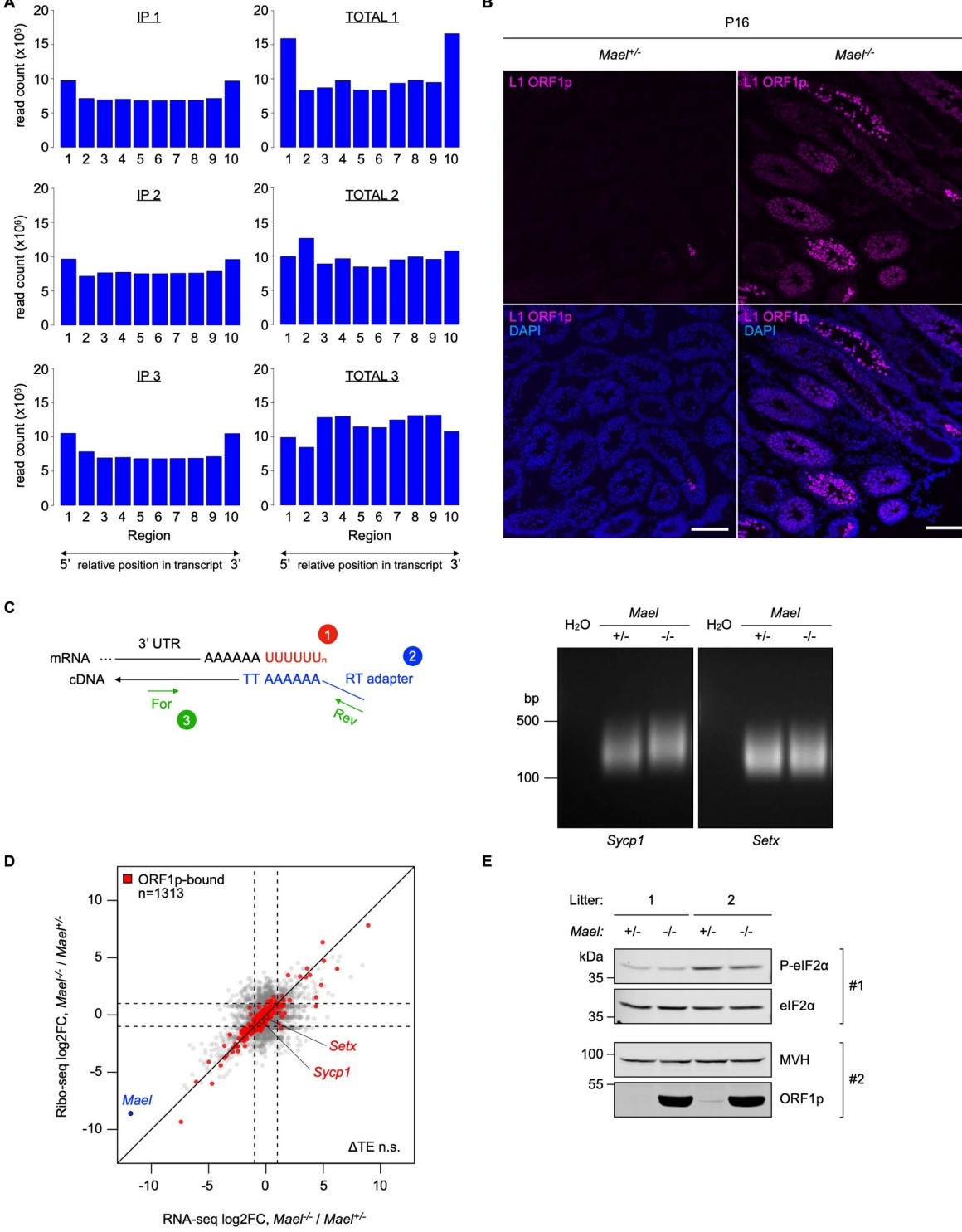

**Fig 6. L1 ORF1p binding to endogenous mRNAs affects neither their stability nor their translation efficiency. (A)** RNA-seq coverage of RNAs detected in anti-L1 ORF1p co-immunoprecipitation samples (IP and TOTAL; shown in triplicate). Each of the segments on the x axis (labeled 1 to 10) corresponds to 10% of transcript lengths; the cumulative number of reads across each segment is plotted (y axis, read count x10$^6$). **(B)** Immunofluorescence staining of L1 ORF1p (magenta) in P16 Mael$^{+/-}$ and Mael$^{-/-}$ testes. Scale bars: 100 μm. **(C)** Analysis of the poly(A) tail of *Sycp1* and *Setx* mRNAs in P16 testes of Mael$^{+/-}$ and Mael$^{-/-}$ littermates. A schematic of the assay is shown on the left; numbers mark the assay steps: 1) poly(U) tailing; 2) cDNA synthesis using an RT adapter; and 3) PCR spanning the poly(A) tail. Representative gels are shown on the right; the assay was performed in biological triplicate. **(D)** Scatterplot showing the relationship

between mRNA level changes (x axis, RNA-seq log2FoldChange) and ribo-footprint changes (y axis, Ribo-seq log2FoldChange) in $Mael^{-/-}$ over $Mael^{+/-}$ testes. A subset of 1313 mRNAs found associated with ORF1p previously in this study are labeled in red. ΔTE n.s.: difference in Translation Efficiency between $Mael^{-/-}$ and $Mael^{+/-}$ non significant. **(E)** Western blot analysis of phosphorylated eIF2α levels (P-eIF2α) in P16 testes of $Mael^{+/-}$ and $Mael^{-/-}$ littermates. Blot #1 and #2 were run and processed in parallel. Blot #1 was first probed with an anti-P-eIF2α antibody, then stripped and re-probed with an anti-eIF2α antibody (pan); blot #2 was probed sequentially with the monoclonal anti-ORF1p (Abcam) and an anti-DDX4/MVH antibody.

To test for the PRKRA effect on L1 mobilization, we performed L1 retrotransposition assays in human HeLa cells following the established protocol [33]. We transfected HeLa cells to co-express mouse PRKRA with a wild-type native mouse L1spa element (pTN201) [20]. To monitor the cytotoxicity of PRKRA overexpression, we co-transfected pPRKRA with a control reporter plasmid (pPAGFP-C1) containing the same selection marker (*neo*) as the L1 retrotransposition plasmid (Fig 7C) [33]. We used the resulting values in the reporter control assay to correct L1 plasmid retrotransposition frequencies in the presence of pEGFP or pPRKRA (see Materials and Methods for details). Retrotransposition frequencies observed in the presence of the control plasmid (pEGFP) in place of pPRKRA were considered the baseline. Fig 7C and 7D demonstrate that PRKRA can enhance the retrotransposition efficiency of a mouse L1spa element in HeLa cells compared to the control. Consistent with ORF1p association with PRKRA in testes and F9 cells, we also observed protein co-localization in HeLa cells (Fig 7E). To confirm that L1 retrotransposition in the presence of pPRKRA proceeds according to the established mechanism, we used plasmids expressing a synthetic mouse wild-type and catalytically inactive L1 elements (S7F Fig) [20,75,76]. To gain insight into the PRKRA mechanism of action in L1 retrotransposition, we measured mouse L1 RNA and ORF1p levels in HeLa cells co-transfected with L1spa and pPRKRA. While L1spa RNA levels decreased by 25% (Fig 7F), its ORF1p levels increased in the presence of PRKRA (Fig 7G and 7H) compared to controls (pEGFP empty vector). Finally, we observed a moderate reduction of eIF2α phosphorylation upon expression of pPRKRA and L1spa in HeLa cells compared to controls (Fig 7I and 7J), consistent with the increase of L1spa ORF1p levels described above (Fig 7G and 7H).

## Discussion

In contrast to the growing understanding of defenses protecting cells and their genomes from mobile DNA, mechanisms of TE endurance remain largely unexplored. In this work, we used an *in vivo* mouse model to characterize L1 interactome in germ cells, the most coveted cell lineage for TE expansion. Prior studies have examined L1 in cultured cells revealing the impressive complexity of L1 interactome (S5 Table) [16,36–39,41,77]. This work demonstrates the conservation of the fundamental principle of L1 association with the multitude of RNA species and RNA granules proteins and offers new insights into ORF1p interactions with host factors.

Our results demonstrate that germ cells overexpressing L1 develop cytoplasmic bodies enriched for ORF1p, a subset of SG, some PB proteins, and the major cytoplasmic deadenylation complex CCR4-NOT. Immunofluorescence and immunogold labeling show that ORF1p accumulates in LBs at exceptionally high levels (this work and [45]). Prior studies in cultured cells have reported L1 association with SGs and their contribution to cellular defenses against L1 [17,36,37,78–80]. Meanwhile, L1 localization to PBs was reported only sporadically [81]. Although LBs harbor numerous RNA granule proteins, particularly those of SGs, we found that LBs contain proteins of both ribosomal subunits. In contrast, SGs and PBs lack fully assembled ribosomes [42,82]. Consistent with ribosome presence in LBs, a significant share of the cellular ORF1p co-sediments with polysomes in sucrose gradients. Ribosomes might

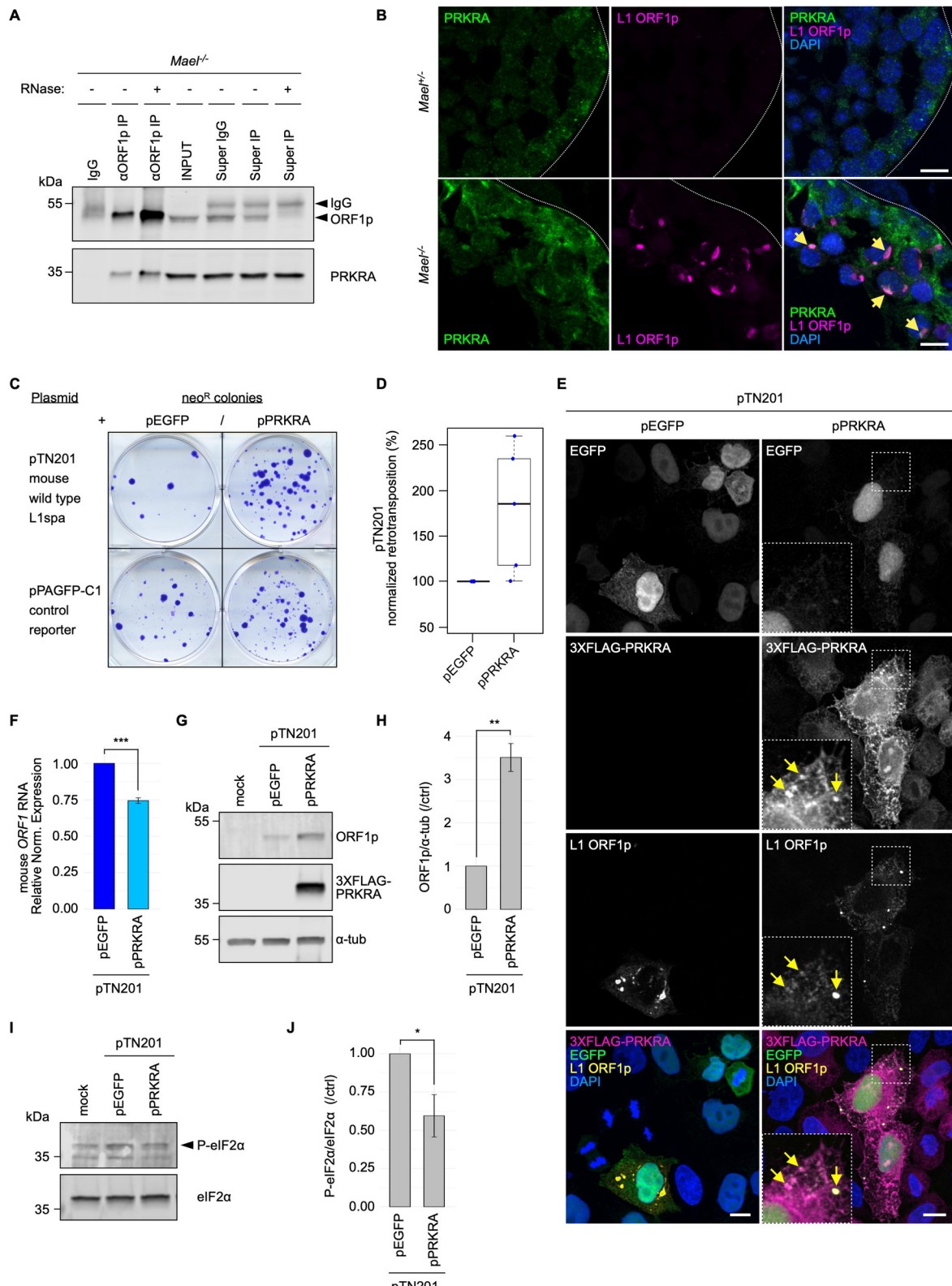

**Fig 7. RNase-resistant ORF1p-interactor PRKRA localizes to LBs *in vivo* and promotes mouse L1 retrotransposition in cultured human cells. (A)** Western blot analysis of PRKRA in anti-L1 ORF1p co-immunoprecipitation samples from *Mael⁻/⁻* testes. Immunoprecipitation with an isotype IgG was used as a negative control; note the higher yield of ORF1p in the presence of RNases. Super:

supernatant. **(B)** Double immunofluorescence staining of PRKRA (green) and L1 ORF1p (magenta) in *Mael*[+/-] and *Mael*[-/-] testes shows PRKRA enrichment in LBs (yellow arrows). Scale bars: 10 μm. **(C)** Representative L1 retrotransposition assay results using the retrotransposition-competent or control plasmids listed on the left, in combination with pEGFP or pPRKRA in HeLa cells. neo[R]: neomycin-resistant colonies. **(D)** Overexpression of mouse PRKRA stimulates retrotransposition of a mouse wild type L1 in HeLa cells. Box-and-whisker plot shows corrected retrotransposition efficiencies from five biological replicates. **(E)** Double immunofluorescence staining of 3XFLAG-PRKRA (anti-FLAG antibody, magenta) and ORF1p (yellow) in HeLa cells co-transfected with pTN201 together with pEGFP or pPRKRA. Boxed areas in pPRKRA images are magnified in the corresponding insets and identify a cell with a moderate expression level of pPRKRA plasmid; exogenous 3XFLAG-PRKRA shows a cytoplasmic distribution that partially overlaps with L1 ORF1p granules (yellow arrows). Scale bars: 10 μm. **(F)** qRT-PCR analysis of mouse L1 RNA (*ORF1* amplicon) derived from pTN201 co-transfected with pEGFP or pPRKRA in HeLa cells. Human *GAPDH* was used as the reference gene. Barplot shows relative normalized expression ± SEM of three biological replicates. Two-tailed unpaired Student's t-test: *** $p \leq 0.001$. **(G)** Western blot analysis of mouse ORF1p derived from pTN201 co-transfected with pEGFP or pPRKRA in HeLa cells. Alpha-tubulin (α-tub) serves as a loading control. **(H)** Barplot showing mean fold change of mouse ORF1p derived from pTN201 co-transfected with pPRKRA in HeLa cells. ORF1p band intensities are normalized to alpha-tubulin (α-tub) and further shown as fold change relative to control (pTN201 + pEGFP). Data are expressed as mean ± SEM of three biological replicates. Two-tailed unpaired Student's t-test: ** $p \leq 0.01$. **(I)** Western blot analysis of human phosphorylated eIF2α (P-eIF2α) in the presence of pTN201 co-transfected with pEGFP or pPRKRA in HeLa cells. eIF2α (pan) serves as a loading control. **(J)** Barplot showing mean fold change of human P-eIF2α in the presence of pTN201 co-transfected with pPRKRA in HeLa cells. P-eIF2α band intensities are normalized to eIF2α and further shown as fold change relative to control (pTN201 + pEGFP). Data are expressed as mean ± SEM of three biological replicates. Two-tailed unpaired Student's t-test: * $p \leq 0.05$. See also S7 Fig.

account for the distinctive appearance of LBs in EM images accentuated by ~25 nm particles, reported neither for SGs nor PBs [83]. LBs are also significantly larger than SGs and PBs and frequently associate with double membranes. Because of the presence of ribosomes, LBs might be structurally or functionally related to neuronal RNA granules [84–86]. Indeed, besides ribosomes, LBs and neuronal granules share several other proteins, including G3BP2, CAPRIN1, FMR1, FXR1, FXR2, ELAV1, ELAV2, PURA, and others [84]. Nonetheless, LBs also differ from neuronal RNA granules, including their sizes and EM appearance, the lack of heavy RNA complexes below polysomes in sucrose gradients, and the association of eIF4A and eIF4E proteins with ORF1p. Thus, LBs are hybrid organelles harboring ORF1p, numerous RNA granule proteins, and ribosomes.

Our data suggest that L1 RNA and ORF1p are crucial for LBs assembly. There is a strong correlation between the increased L1 expression levels and the formation of LBs during meiotic prophase I ([45] and this study). Indeed, *Mael*[-/-] mice on the 129SvJae genetic background (as opposed to B6 used in this work) express lower ORF1p levels and form LBs less frequently [50]. Furthermore, small LB precursors are evident in wild-type germ cells expressing low levels of L1 in early meiosis (this work and [51]).

Our data also suggest that ORF1p drives LB assembly by interacting with RNAs and thus attracting numerous RNA-binding proteins. Consistent with prior findings [29,30], ORF1p shows a preference for RNA of evolutionarily young L1MdT and L1MdA elements. Unexpectedly, we observed no enrichment for SINE B1 and B2 RNAs, known to be mobilized by the L1 machinery [64,65,87,88], and strong enrichment for MMERVK10C-int RNA [89]. We also found that ORF1p is associated with numerous genic mRNAs, and such interactions correlate with mRNA expression levels. These observations agree with the evidence of the non-essential role of ORF1p in SINE retrotransposition but its requirement for generating processed pseudogenes [65,90]. Interestingly, the pool of heterogeneous RNAs associating with ORF1p shares the capacity for translation. Considering that ORF1p interacts with ribosomes, as shown by orthogonal approaches, we propose that ORF1p might interact with ribosome-associated RNAs. Notably, our observations *in vivo* agree with recent reports of ORF1p capacity to phase separate and form condensates with RNAs that are required for L1 retrotransposition, further supporting the leading role of ORF1p in LBs formation [91,92].

Prior studies proposed that L1 RNP localization to SGs might restrict L1 activity as a defense mechanism [41]. Indeed, the colocalization and RNA-mediated interactions of ORF1p with various RNA granule proteins identified in this study might suggest that LBs are sites of RNA storage, destabilization, or translational repression. However, a paradoxical feature of LBs is that despite numerous proteins implicated in RNA regulation, we find no evidence of reduced RNA stability and translation. For example, although the CCR4-NOT complex is a major deadenylase, we did not observe augmented deadenylation of ORF1p-associated RNAs. These observations suggest that the CCR4-NOT complex might be inactivated in the context of LBs by other factors, such as shown for RNF219 E3-ubiquitin ligase [93]. Alternatively, the CCR4-NOT complex might perform a different function in LBs, such as facilitating the co-translational assembly of ORF1p trimers, which must rapidly oligomerize to ensure L1 retro-transposition [94]. This idea reflects a previously established role of CCR4-NOT in the assembly of the proteasome, RNA Polymerase II, and the SAGA complex [95–97]. Furthermore, LBs contain ribosomes, and ORF1p interacts with ribosomal proteins following the stringent RNase treatment. Indeed, we did not detect reduced translation of mRNAs bound by ORF1p and the transcriptome in general. This result can be explained by the relocalization of only a minor fraction of germ cell mRNAs to LBs, rendering any translational changes difficult to detect overall. Alternatively, LBs might represent sites of active translation rather than mRNA storage or degradation. In agreement with this hypothesis, *Setx*, the most enriched mRNA in ORF1p complexes localizes to LBs and shows no reduction in translational efficiency or signs of deadenylation.

Further support for the above hypothesis comes from the evidence of ORF1p adapting PRKRA, a PKR factor, to prevent the activation of a crucial arm of the integrated stress response. Although under some conditions PRKRA (or its human homolog PACT/RAX) inhibits PKR [71,74,98,99], it primarily functions as a PKR activator, even in the absence of dsRNA [100]. Our data suggest that ORF1p-PKRKA interactions preclude PKR activation, phosphorylation of eIF2α, and translational inhibition. Such negative control could happen either through ORF1p sequestering PRKRA and preventing it from activating PKR, or actively inducing PRKRA to inhibit PKR. ORF1p might fulfill the role of TRBP/TARBP2 protein that prevents PKR activation by heterodimerizing with PRKRA [99]. Under stress the TRBP-PRKRA complex disassembles, allowing PRKRA to activate PKR. ORF1p may bind the released PRKRA, thus preventing PKR activation and translational inhibition. In support of this hypothesis, ORF1p complexes characterized in this study contained only small amounts of TARBP2 compared to PRKRA, suggesting a preferential binding of ORF1p to PRKRA in place of TARBP2. Notably, PKR is dispensable for L1 RNP localization to SGs in cultured cells devoid of three major eIF2α activating kinases, including PKR [41]. Furthermore, we observed enhancement of L1spa retrotransposition in HeLa cells upon PRKRA overexpression. This effect can be partially explained by the increase in ORF1p levels in the presence of PRKRA, possibly facilitating L1 mobilization. Further experiments will be needed to address the mechanism of ORF1p augmentation, which can happen either at the translational level through a PRKRA-mediated translational stimulation, or post-translation through protein stabilization.

Taken together our data suggest that—like viruses—L1 encodes an active mechanism that counteracts cellular defenses. Viruses have evolved many countermeasures to overcome translational inhibition [101]. A particularly successful strategy employed by viruses is to inhibit the initiation of cap-dependent translation. This strategy allows viruses to attract ribosomes to viral RNAs and increase translation initiation via Internal Ribosomal Entry Site elements. The strategy, however, is likely unfavorable for TEs since the host's fitness ensures TE propagation [23]. Therefore, the proposed mechanism of relieving translational inhibition could be the optimal solution that satisfies the needs of L1 and the host.

To conclude, this study revealed the formation of a novel cytoplasmic RNA body upon L1 overexpression in mouse germ cells. LBs are hybrid structures, harboring RNA metabolism and translational repression proteins and ribosomes. Despite the nucleation of LBs, no successful containment of L1 occurs, and the stability and translation efficiency of cellular RNAs are not compromised. In addition, L1 appears to evade cellular defenses and translational shut-off by co-opting the PKR-associated protein PRKRA. In this manner, L1 might increase its chances of further spreading in the host genome. We speculate that the described mechanism might be of particular significance in somatic cells lacking piRNAs naturally and might account for the emerging roles of L1 in aging and disease [102–105]. Additional studies will provide deeper mechanistic insights into LBs functions and L1-PRKRA/PKR interactions.

## Materials and methods

### Ethics statement

All procedures involving mice were performed according to ethical regulations and were approved by the Institutional Animal Care and Use Committee of the Carnegie Institution for Science.

### Mice and testis samples preparation

C57BL/6 Maelstrom (*Mael*) mutant mice were previously generated [45]. C57BL/6 WT, *Mael*$^{+/-}$ and *Mael*$^{-/-}$ male mice were euthanized at ~ 3 months of age, testes were dissected and the tunica albuginea was removed in ice cold phosphate-buffered saline (PBS; 137 mM NaCl, 2.7 mM KCl, 10 mM $Na_2HPO_4$, 1.8 mM $KH_2PO_4$, pH 7.4).

For electron microscopy (EM), testes were fixed and processed as described in [50]. For immunofluorescence and RNA Fluorescence In Situ Hybridization (RNA FISH), testes were fixed in freshly prepared 4% paraformaldehyde (PFA, EM grade, Electron Microscopy Sciences) in PBS for 4 hours on ice and further equilibrated in a series of sucrose solutions at increasing concentrations (10, 20 and 30% sucrose in PBS) at 4˚C. Following embedding in OCT compound (Tissue-Tek), testes were sectioned into 10 μm slices on poly-L-lysine-coated slides. Slides were immediately stored at -80˚C.

### Antibodies

The following primary antibodies and dilutions were used: rabbit polyclonal anti-ORF1p (full length protein; [106]) diluted 1:10000 for western blot and 1:1000–1:3000 for immunofluorescence; rabbit monoclonal anti-ORF1p (Abcam, ab216324) diluted 1:1500 for western blot and 1:1000 for immunofluorescence; rabbit polyclonal anti-ribosomal protein S6/RPS6 (Novus, NB100-1595) diluted 1:1200 for western blot and 1:20 for immunofluorescence; rabbit polyclonal anti-ribosomal protein L28/RPL28 (Abcam, ab254927) diluted 1:100 for immunofluorescence; rabbit polyclonal anti-DDX6 (Bethyl, A300-460A-M) diluted 1:15 for immunofluorescence; rabbit monoclonal anti-CNOT7 (Abcam, ab195587) diluted 1:1000 for immunofluorescence; rabbit polyclonal anti-CNOT1 (Proteintech, 14276-1-AP) diluted 1:200 for immunofluorescence; mouse monoclonal anti-DCP1A (3G4, Novus, H00055802-M06) diluted 1:100 for immunofluorescence; rabbit polyclonal anti-PRKRA (Abcam, ab31967) diluted 1:1000 for western blot; rabbit polyclonal anti-PRKRA (Epigentek, A72895) diluted 1:300 for immunofluorescence; mouse monoclonal anti-FLAG M2 (Sigma, F1804) diluted 1:1000 for western blot and 1:500 for immunofluorescence; rabbit monoclonal anti-phospho-eIF2α (Ser51) (Abcam, ab32157) diluted 1:2000 for western blot on HeLa cell extracts; rabbit monoclonal anti-phospho-eIF2α (Ser51) (Cell Signaling, 3398S) diluted 1:1000 for western blot on mouse testis extracts; rabbit polyclonal anti-eIF2α (pan) (Cell Signaling, 9722S) diluted

1:1000 for western blot; rabbit polyclonal anti-DDX4/MVH (Abcam, ab13840) diluted 1:3000 for western blot; mouse monoclonal anti-α-tubulin (Sigma, T6199) diluted 1:10000 for western blot.

The following secondary antibodies and dilutions were used: Alexa Fluor 488 donkey anti-rabbit IgG (Invitrogen, A21206), Alexa Fluor 568 donkey anti-rabbit IgG (Invitrogen, A10042), Alexa Fluor 647 donkey anti-rabbit IgG (Invitrogen, A31573) and Alexa Fluor 647 donkey anti-mouse IgG (Invitrogen, A31571) diluted 1:500 for immunofluorescence; IRDye 800CW goat anti-rabbit (LI-COR, 926–32211) and IRDye 680RD goat anti-rabbit (LI-COR, 926–68071) diluted 1:10000 for western blot.

## Plasmids

The control vector pEGFP was generated by Gibson assembly [107] of the following DNA fragments: the EF-1 alpha promoter (amplified from pEF1a-mRor2WT, Addgene #2261, a gift from Roel Nusse [108]) and the EGFP-ori-AmpR fragment (amplified from pCMV_ABE-max_P2A_GFP, Addgene #112101, a gift from David Liu [109]).

The pPRKRA plasmid, expressing N-terminal 3XFLAG-tagged PRKRA (NCBI Reference Sequence: NM_011871) mouse protein, was generated by Gibson assembly of the following fragments: the EF-1 alpha promoter (from Addgene #2261), the Kozak-3XFLAG fragment (from c3GIC9, Addgene #62191, a gift from Lukas Dow [110]), the PRKRA CDS (amplified from a *Mael*$^{-/-}$ testis cDNA sample) and the P2A-EGFP-ori-AmpR fragment (from Addgene #112101).

pTN201, pCEPsmL1 and pCEPsmL1mut plasmids have been described elsewhere [20,76] and were kindly provided by Jeffrey Han (Tulane University); pPAGFP-C1 plasmid (Addgene #11910) was a gift from Svetlana Deryusheva (Carnegie Institution for Science, Embryology).

## Cell culture and DNA transfection

Human HeLa cells (kindly provided by Kamena Kostova, Carnegie Institution for Science, Embryology) and mouse F9 carcinoma cells (ATCC CRL-1720) were maintained in DMEM (Gibco, 11960051) supplemented with 10% FBS, 1X Glutamax (Gibco, 35050061) and 1X penicillin/streptomycin (Gibco, 15140122). Coating of culture surfaces with 0.1% gelatin (Sigma, G1393) was performed to improve F9 cells adhesion. Cell lines were routinely tested for Mycoplasma contamination using the Venor GeM Mycoplasma Detection Kit (Sigma, MP0025).

HeLa cells were transfected using Fugene HD reagent (Promega, E2311) at a 3:1 Transfection Reagent:DNA ratio. Cells were harvested for RNA or protein analyses 72 hours after transfection. Fugene HD only-treated cells (mock) were used as a negative control.

F9 cells were transfected using Lipofectamine STEM (Invitrogen, STEM00003). 48 hours after transfection, cells were either fixed for immunofluorescence assays or harvested for differential cytoplasmic / nuclear protein extraction or co-immunoprecipitation. Lipofectamine only-treated cells (mock) were used as a negative control in immunofluorescence assays.

## Immunofluorescence

For immunofluorescence on testis sections, slides were thawed and left to dry at 42°C for 5 minutes. Sections were then rehydrated in PBS for 10 minutes at room temperature before permeabilization.

For immunofluorescence on F9 cultured cells, cells were grown on fibronectin-coated coverslips (Corning, 354088) in 6-well plates and transfected as described above. 48 hours after transfection, cells were fixed in 4% PFA in PBS for 12 minutes at room temperature and further rinsed in PBS before permeabilization.

Samples were permeabilized with 0.1% Triton X-100 in PBS for 10 minutes and blocked with 10% normal serum / 0.05% Triton X-100 in PBS for 1 hour at room temperature. Samples were then probed with the appropriate primary antibody, diluted in blocking solution with 3% normal serum, overnight at 4˚C. Slides were washed once with 0.1% Triton X-100 and twice with 0.05% Triton X-100 in PBS and further incubated with the corresponding Alexa Fluor-conjugated secondary antibody diluted in blocking solution with 3% normal serum, for 1 hour and 30 minutes at room temperature. Slides were washed as above, counterstained with DAPI and mounted with ProLong Diamond Antifade Mountant (Thermo Fisher, P36961).

For immunostaining performed using two primary antibodies derived from the same species (rabbit anti-ORF1p with either rabbit anti-RPS6, rabbit anti-RPL28, rabbit anti-CNOT7, rabbit anti-DDX6, rabbit anti-CNOT1, or rabbit anti-PRKRA antibodies), samples were probed as described above using one of the two rabbit primary antibodies. Samples were then washed, briefly fixed with 4% PFA in PBS and probed again with the other rabbit primary antibody directly labeled using either the Zenon rabbit IgG labeling kits (Invitrogen, Z25302 or Z25306), or the Mix-n-Stain CF647 Antibody labeling kit (Sigma, MX647S100), according to manufacturer's instructions. To avoid cross-talk between channels, combinations of Alexa Fluor 488 with either red (568) or far red (647) Alexa Fluor fluorophores were used.

For DCP1A and ORF1p double labeling, the mouse anti-DCP1A antibody was covalently conjugated to a 488 fluorophore using the Mix-n-Stain CF488A Antibody labeling kit (Sigma, MX488AS100) and used together with the rabbit polyclonal anti-ORF1p antibody covalently conjugated to a 647 fluorophore with the Mix-n-Stain CF647 Antibody labeling kit. This strategy eliminates the background staining generated by the use of an anti-mouse secondary antibody on mouse tissues.

## Hybridization Chain Reaction (HCR) RNA FISH

HCR RNA FISH (v3.0) using split-initiator probes was performed using custom DNA probe sets, DNA HCR hairpin amplifiers, Probe Hybridization and Amplification buffers purchased from Molecular Instruments Inc. (Los Angeles, CA, USA) as described in [58] with modifications as follows. Mouse testis slides were thawed and left to dry at 42˚C for 5 minutes. Sections were rehydrated in PBS and permeabilized in 0.2% Triton X-100 in PBS for 10 minutes at room temperature. Slides were then incubated with 1 ml pre-warmed (37˚C) pepsinization solution (0.005% pepsin in 0.01 N HCl) for 6 minutes at room temperature to unmask the RNA. Pepsin was inactivated with 50 mM $MgCl_2$ in PBS and sections were post-fixed in 1% PFA in PBS for 10 minutes at room temperature. For RNase-treated samples, slides were further incubated with 0.2 μg/μl RNase A in PBS for 1 hour at 37˚C, before proceeding with the following steps.

Slides were washed twice in 2X SSC and incubated in 50% formamide in 2X SSC for 2 hours at room temperature in a humidified chamber. The humidified chamber was then transferred to 37˚C and pre-warmed for one additional hour. Slides were pre-hybridized in Probe Hybridization buffer (Molecular Instruments Inc.) for 30 minutes at 37˚C, then probe solutions were added and incubated overnight at 37˚C. Setx and U6 snRNA probes were used at a final concentration of 5 nM; probes directed to MMERVK10c-int and L1spa were used at a final concentration of 2.5 nM in Probe Hybridization buffer. For background estimation, probes were omitted in negative control slides. Slides were immersed in 2X SSC for 20 minutes at room temperature to remove excess probes, then washed with 0.1% Tween-20 in 0.2X SSC (multiplexed MMERVK10c-int and L1spa slides) or with 0.1% Tween-20 in 1X SSC (U6 slides, and multiplexed Setx and L1spa slides). Slides were then incubated in Amplification buffer (Molecular Instruments Inc.) for 30 minutes at room temperature before proceeding to probes amplification using snap-cooled fluorescent HCR hairpins diluted in Amplification buffer,

overnight at room temperature. Hairpins directed to Setx probes (B1 amplifier) were conjugated with Alexa 488 (B1-488); hairpins directed to U6 probes (B4 amplifier) were conjugated with Alexa 488 (B4-488); hairpins directed to MMERVK10c-int probes (B2 amplifier) were conjugated with Alexa 546 (B2-546); hairpins directed to L1spa probes (B4 amplifier) were conjugated with Alexa 647 (B4-647). Slides were immersed in 2X SSC for 15 minutes at room temperature to remove excess hairpins and then further washed with 0.1% Tween-20 in 1X SSC, four times at room temperature. Samples were counterstained with DAPI and mounted with ProLong Diamond Antifade Mountant (Thermo Fisher, P36961).

## Combined immunofluorescence and HCR RNA FISH

Testis sections (Fig 1G) were first probed for L1 ORF1p (rabbit polyclonal antibody), according to the immunofluorescence protocol described previously. After the secondary antibody incubation (Alexa Fluor 488 donkey anti-rabbit IgG; Invitrogen, A21206) and the subsequent washes in PBS / Triton X-100, slides were washed twice in 2X SSC, and then transferred in 50% formamide in 2X SSC. The following FISH steps (pre-hybridization, hybridization with probes directed to L1spa, and signal amplification with Alexa 647-conjugated hairpins) were performed as described previously in the HCR RNA FISH section; all washes were performed with 0.1% Tween-20 in 1X SSC.

## Microscopy and image analysis

Following immunofluorescence labeling or RNA FISH, samples were imaged using a Leica SP5 confocal laser scanning microscope or a Leica Stellaris 8 FALCON confocal microscope with LAS X software. For colocalization analyses by two-color FISH or immunofluorescence, images were acquired with settings adjusted to avoid fluorescence bleed-through and channels were scanned sequentially to obtain complete separation of the signals. Image processing was performed using ImageJ (https://imagej.nih.gov/ij/) or with the Imaris software (Oxford Instruments).

## Protein extraction

Proteins from transfected HeLa cells were extracted with RIPA buffer containing 50 mM Tris-HCl pH 8.0, 150 mM NaCl, 1% NP-40, 0.5% sodium deoxycholate, Halt protease and phosphatase inhibitors (Thermo Fisher, 78445) in water. Cells were incubated for 30 minutes on ice and lysates were centrifuged at 16,000 x $g$ for 10 minutes at 4°C. Supernatants were recovered and proteins were quantified using the BCA assay kit (Thermo Scientific, 23225).

Differential cytoplasmic / nuclear protein extracts from transfected F9 cells were prepared using the NE-PER nuclear and cytoplasmic extraction reagents (Thermo Scientific, 78833), according to manufacturer's instructions.

## Sucrose gradients ultracentrifugation

Testes from C57BL/6 $Mael^{+/-}$ and $Mael^{-/-}$ mice (~ 3 months of age) were dissected as previously described. For treatment with cycloheximide (CHX, Sigma-Aldrich, C4859), freshly dissected testes were transferred in ice cold PBS containing 200 μg/ml CHX and they were further homogenized in Lysis buffer containing 100 mM KCl, 20 mM Tris-HCl pH 7.5, 5 mM $MgCl_2$, 0.3% NP-40, 200 μg/ml CHX, Halt protease and phosphatase inhibitors (Thermo Fisher, 78445), RNasin Ribonuclease Inhibitor (Promega, N2515) in nuclease-free water. For treatment with EDTA, testes were homogenized in Lysis buffer containing 100 mM KCl, 20 mM Tris-HCl pH 7.5, 2 mM $MgCl_2$, 0.3% NP-40, Halt protease and phosphatase inhibitors (Thermo Fisher, 78445), RNasin Ribonuclease Inhibitor (Promega, N2515) in nuclease-free water.

Testes were homogenized using a Kontes motorized pestle and obtained lysates were incubated 20 minutes on ice. Lysates were centrifuged at 12,000 x $g$ for 10 minutes at 4˚C and supernatants were recovered. For treatment with EDTA, EDTA (Invitrogen, AM9260G) was added to the supernatant at this point, at a final concentration of 10 mM.

RNA concentration was quantified using a Qubit Fluorometer (Thermo Fisher Scientific) and equal amounts of RNA (100 µg) in the same final volume (~300 µl extract) were loaded onto linear 10–50% (w/v) sucrose gradients. Gradients were centrifuged in a SW41 rotor (Beckman) at 35,000 rpm for 2 hours and 30 minutes at 4˚C. For CHX gradients, sucrose solutions were prepared in 100 mM KCl, 20 mM Tris-HCl pH 7.5, 5 mM $MgCl_2$ in nuclease-free water. For EDTA gradients, sucrose was dissolved in 40 mM KCl, 25 mM Tris-HCl pH 7.5, 10 mM EDTA [57] in nuclease-free water. Sucrose gradients were fractionated using a Gradient Master station (Biocomp; 12 fractions of 1 ml each), while absorbance (254 nm) was continuously recorded to generate a UV profile of the gradient.

For treatment with RNase, samples were prepared following the EDTA treatment method described above, omitting RNasin Ribonuclease Inhibitor from the Lysis buffer. Before layering onto EDTA gradients, samples were incubated with 10 µg/ml RNase A (Invitrogen, AM2270) and 25 U/ml RNase T1 (Thermo Fisher, EN0541) for 30 minutes at room temperature with gentle mixing.

### Protein precipitation

Proteins were precipitated from sucrose fractions using trichloroacetic acid (TCA; Sigma-Aldrich, T6399). Briefly, 250 µl of 50% (w/v) TCA in water were added to each 1 ml sucrose fraction, in the presence of 0.016% sodium deoxycholate as a carrier. Samples were incubated on ice for 30 minutes and then centrifuged at 16,000 x $g$ for 30 minutes at 4˚C. Protein pellets were washed twice with 1 ml ice-cold acetone, resuspended in 60 µl of SDS sample buffer (2% sodium dodecyl sulfate (SDS), 50 mM Tris-HCl pH 6.8, 10% glycerol, 100 mM dithiothreitol (DTT), 0.003% bromophenol blue in water) and resolved on 12% SDS-polyacrylamide gels.

### Western blot

Proteins were resolved on SDS-polyacrylamide gels and transferred onto PVDF Immobilon-FL membranes (Millipore, IPFL00010). Membranes were blocked with Intercept blocking buffer in PBS (LI-COR, 927–70001) or with 3% Bovine Serum Albumin (BSA) in Tris-buffered saline (TBS; 150 mM NaCl, 20 mM Tris-HCl, pH 7.6) for phospho-eIF2α detection, for one hour at room temperature. Membranes were probed with primary antibodies diluted in Intercept blocking buffer containing 0.1% Tween-20 or with anti-phospo-eIF2α antibodies diluted in 1% BSA in TBS, overnight at 4˚C. Membranes were washed three times with PBS or TBS containing 0.1% Tween-20 and further incubated with IRDye-conjugated secondary antibodies (LI-COR) diluted in the same buffer used for the corresponding primary antibody, for one hour at room temperature. Membranes were washed three times and signals were detected with the Odyssey CLx imaging system (LI-COR).

Signals were quantified with Image Studio software v5.2.5 (LI-COR). The stripping buffer used for blot #1 in Fig 6E is NewBlot PVDF Stripping buffer (LI-COR, 928–40032), according to manufacturer's guidelines.

### RNA extraction

RNA from 500 µl of each sucrose fraction or from a fraction (5%) of the gradient input was extracted by adding 500 µl of TRIzol reagent (Invitrogen) followed by 200 µl of chloroform. Samples were incubated at room temperature for 10 minutes with gentle agitation and then

centrifuged at 12,000 x $g$ for 15 minutes at 4˚C. Supernatants were collected and RNA was precipitated with 1 volume of isopropanol and 1 μl of GlycoBlue (Invitrogen, AM9515). RNA was recovered by centrifugation at 12,000 x $g$ for 30 minutes at 4˚C. RNA pellets were washed with ice-cold 75% ethanol in nuclease-free water and further resuspended in 11 μl of nuclease-free water. 5 μl were used for DNase treatment with TURBO DNase (Invitrogen, AM2238) in 10 μl final volume and further processed for cDNA synthesis.

RNA from transfected HeLa cells was extracted with 500 μl of TRIzol reagent (Invitrogen) and further purified with Direct-zol RNA Miniprep kit (Zymo research, R2050), according to manufacturer's protocol. 2 μg per sample were used for DNase treatment with TURBO DNase (Invitrogen) and further processed for cDNA synthesis.

## cDNA synthesis and qRT-PCR

DNase-treated RNA was divided in 2 samples of equal volume (+RT and -RT) and processed for cDNA synthesis using an oligo(dT)$_{20}$ and the SuperScript III first-strand synthesis system (Invitrogen, 18080051), according to manufacturer's instructions. The reverse transcriptase was omitted in -RT samples.

cDNA samples obtained from sucrose fractions were diluted 1:35 in nuclease-free water, while samples obtained from gradient inputs were diluted 1:100; cDNA samples from human HeLa cells were diluted 1:80 in nuclease-free water. 4 μl of diluted cDNAs were added to each PCR reaction containing the SsoAdvanced Universal SYBR Green Supermix (Bio-Rad, 1725271) and primers at a final concentration of 300 nM each, in a final volume of 10 μl. The following primer couples were used: ORF1-For: ATG GCG AAA GGT AAA CGG AG, ORF1-Rev: AGT CCT TCT TGA TGT CCT CT (for mouse L1 ORF1); mActb-For: CGG TTC CGA TGC CCT GAG GCT CTT, mActb-Rev: CGT CAC ACT TCA TGA TGG AAT TGA (for mouse beta-actin); hGAPDH-For: AAT CCC ATC ACC ATC TTC CAG, hGAPDH-Rev: AAA TGA GCC CCA GCC TTC (for human GAPDH).

Mean (n = 3) Ct (threshold cycle) values were used to quantify mRNA levels. The distribution of ORF1 and beta-actin mRNAs in sucrose gradients was determined using the ΔCt method. The Ct value of the gradient input was set as the reference and obtained values were further converted into percentages. Relative quantities of mouse ORF1 in transfected HeLa cells were analyzed using the ΔΔCt method, with human GAPDH as the reference gene.

## Co-immunoprecipitation

Proteins from testes and from transfected F9 cells were extracted using the same Lysis buffer described previously for sucrose gradients ultracentrifugation (treatment with EDTA). Protein concentrations were determined with BCA assay (Thermo Scientific, 23225). For RNase treatment before co-immunoprecipitation, RNA concentration in extracts was determined using a Qubit Fluorometer (Thermo Fisher Scientific) and samples containing approximately 50 μg RNA were incubated with 10 μg/ml RNase A (Invitrogen, AM2270), 25 U/ml RNase T1 (Thermo Fisher, EN0541) and 50 U/ml RNase I (Lucigen, N6901K) for 30 minutes at room temperature with gentle mixing. In these cases, RNasin Ribonuclease Inhibitor was omitted from the Lysis buffer.

For co-immunoprecipitation followed by RNA-seq (Fig 3A–3D), pools of sucrose fractions 5, 6, 7 and 8—obtained from $Mael^{-/-}$ testicular extracts (TOTAL) fractionated in the presence of EDTA—were used as input (INPUT) for either anti-ORF1p immunoprecipitations (IP) or incubations with beads only (BO).

Rabbit monoclonal anti-ORF1p antibody (Abcam, ab216324) or rabbit IgG isotype (Abcam, ab172730) were coupled to Dynabeads M-270 Epoxy, using the Dynabeads Antibody

Coupling kit (Invitrogen, 14311D) at a ratio of 10 µg ligand per mg of beads. For BO samples, the ligand was substituted with an equivalent volume of buffer C1 of the kit. 1.5–2 mg of beads were added to 1.5–2 mg of proteins or to pools of 5–8 fractions and incubated for 2 hours and 30 minutes at 4˚C. Beads were washed four times in 100 mM KCl, 20 mM Tris-HCl pH 8.0, 0.3% NP-40, Halt protease and phosphatase inhibitors in nuclease-free water, for 5 minutes at 4˚C with rotation. For samples washed at a higher stringency, beads were washed twice as above and twice in the same buffer with a higher concentration of KCl (300 mM). After the last wash, samples were transferred to clean tubes. For western blot and silver staining in poly-acrylamide gels (SilverQuest staining kit; Invitrogen, 45–1001), precipitated proteins were eluted with 0.2 M Glycine pH 2.6. For RNA-seq, RNA was eluted by adding TRIzol reagent to Dynabeads and extracted as previously described.

## Mass spectrometry and data analysis

For LC/MS/MS analyses, elution of immunoprecipitated proteins was performed at Poochon Scientific (Frederick, MD, USA) as follows. Dynabeads were dissolved in 2% SDS, 25 mM $NH_4HCO_3$, 10 mM DTT in water. Samples were denatured at 95˚C and then processed for in-solution trypsin digestion. The digested peptide mixture was then concentrated and desalted. Desalted peptides were reconstituted in 20 µl of 0.1% formic acid, of which 16 µl were analyzed by LC/MS/MS.

The LC/MS/MS analysis of samples was carried out using a Thermo Scientific Q-Exactive Hybrid Quadrupole-Orbitrap Mass Spectrometer and a Thermo Dionex UltiMate 3000 RSLCnano System. Peptide mixtures from each sample were loaded onto a peptide trap cartridge at a flow rate of 5 µL/min. The trapped peptides were eluted onto a reversed-phase Pico-Frit column (New Objective, MA) using a linear gradient of acetonitrile (3–36%) in 0.1% formic acid. Eluted peptides from the PicoFrit column were ionized and sprayed into the mass spectrometer, using a Nanospray Flex Ion Source ES071 (Thermo) under the following settings: spray voltage 1.8 kV; capillary temperature 250˚C. Other settings were empirically determined.

The raw data files acquired from each sample were searched against mouse protein sequences database using the Proteome Discoverer 1.4 software (Thermo Scientific) based on the SEQUEST algorithm. The Carbamidomethylation (+57.021 Da) of cysteines was set as a fixed modification, whereas Oxidation Met and Deamidation Q/N-deamidated (+0.98402 Da) were set as dynamic modifications. The minimum peptide length was specified to be 5 amino acids. The precursor mass tolerance was set to 15 ppm, whereas fragment mass tolerance was set to 0.05 Da. The maximum false peptide discovery rate was specified as 0.01. The resulting Proteome Discoverer Report contains all assembled proteins, with Peptide Spectrum Match (PSM) counts.

PSM counts were used to determine the abundance of the corresponding protein in each eluate. For background subtraction based on PSM counts in negative controls (BO or IgG iso-type), proteins showing the same or higher counts in negative controls than corresponding IP samples were excluded from subsequent analyses. Furthermore, proteins with PSM counts $\geq$ 10 in negative controls and with a difference < 3 between IP samples and negative controls were also excluded. For the evaluation of behavior at higher stringency conditions, proteins decreased more than three times or completely lost upon higher stringency washes (300 mM KCl) were filtered out. Only proteins common to all replicates were considered for further analyses. This strategy returned n = 80 high confidence ORF1p interactors.

The relative abundance of the identified proteins in each eluate was then calculated by adjusting the PSM counts to molecular weights (MW in kDa) [111]. To allow a sample-to-

sample comparison of abundances, the obtained PSM / MW ratios were further normalized to the ratio obtained for the bait (L1 ORF1p). Physical and functional interactions were retrieved using the STRING database v11.5 (https://string-db.org/; [67]).

## RNA-seq analysis

A total of 200 ng RNA (for TOTAL and INPUT samples), 20–40 ng RNA (for IP samples) and the RNA yield in the same volume of corresponding IP samples (for BO samples) were treated with TURBO DNase (Invitrogen, AM1907) in 10 μl final volume. Libraries were generated using the TruSeq Stranded Total RNA kit (Illumina) with Ribo-Zero rRNA depletion. 75 bp single-end reads were sequenced on an Illumina NextSeq 500 system.

For genomic repeats analysis, reads were mapped to mouse mm10 reference genome using STAR v2.6.0c [112], option—outFilterMultimapNmax 100. The obtained .bam file was used as input for Telescope v1.0.3 [113] to reassign ambiguously mapped reads. The updated counts were collapsed to single repeat subfamilies using a custom Perl script and normalized using EBSeq v1.26.0 [114] in R.

For coding genes analysis, reads were mapped to mouse mm10 using STAR v2.6.0c. Read counts were normalized using DESeq2 v1.24.0 [115] in R. Genes were considered significatively enriched with a log2 Fold Change > 1 and a padj < 0.05. Transcript lengths (including UTRs and CDS) for n = 2081 L1 ORF1p-bound mRNAs were retrieved from Ensembl using the biomaRt package in R. The max transcript length value was used for the correlation analysis in Fig 3D. Gene ontology analysis was performed using DAVID Bioinformatics Resources v6.8.

## Ribosome profiling

Lysates were generated from postnatal day 16 (P16) C57BL/6 *Mael*[+/-] and *Mael*[-/-] testes. Testes were homogenized in Lysis buffer [116] supplemented with Halt protease and phosphatase inhibitors (Thermo Fisher, 78445) and Murine RNase inhibitor (New England Biolabs, M0314S), using a Kontes motorized pestle. Lysates were clarified by centrifugation at 12,000 x *g* for 10 minutes at 4˚C. Supernatants were recovered and the RNA concentration was determined using a Qubit Fluorometer (Thermo Fisher Scientific). To generate monosomes, 20 μg of RNA per sample were incubated with 5U MNase (Takara Bio, 2910A) supplemented with 5 mM $CaCl_2$ in 200 μl final volume, for 1 hour at room temperature with gentle agitation. Reactions were quenched with 6.25 mM EGTA. Digested extracts were then loaded on 1 M sucrose cushions [116] and centrifuged at 70,000 rpm for 2 hours at 4˚C in a TLA100.4 rotor (Beckman). Ribosome pellets were resuspended in 300 μl of TRIzol reagent (Invitrogen). At this point, aliquots of testicular extracts containing 2.5 μg of total RNA were also resuspended in 300 μl of TRIzol to recover RNA for the corresponding RNA-seq analysis. All samples were purified using the Direct-zol RNA Miniprep kit (Zymo Research, R2050), including the in-column DNase I treatment.

Total RNA samples were further processed to generate libraries using the TruSeq Stranded Total RNA kit (Illumina) with Ribo-Zero rRNA depletion. RNA purified from ribosome pellets was instead processed for ribosome footprints purification (28–34 nt) and libraries generation, as described in [116], with the following modifications. Our 5' adenylated linker sequence is: /5rApp/NNN NNA TCG AGA TCG GAA GAG CAC ACG TCT GAA CTC/ 3ddC/ [117]. Footprints were reverse transcribed with the SuperScript III first-strand synthesis system (Invitrogen, 18080051) for 40 minutes at 55˚C, using the following reverse transcription primer: /5Phos/AGA TCG GAA GAG CGT CGT GTA GGG AAA GAG/iSp18/CTG GAG TTC AGA CGT GTG [117].

75 bp single-end reads were sequenced on an Illumina NextSeq 500 system. Ribo-seq reads were preprocessed with FASTX-Toolkit v0.0.14 (http://hannonlab.cshl.edu/fastx_toolkit/) to clip the linker sequence (ATC GAG ATC GGA AGA GCA CAC GTC TGA ACT C) and trim the UMI (5Ns) from the 3' end. Clipped Ribo-seq reads were aligned to mouse ribosomal DNA (Genbank: BK000964.1) using Bowtie v1.2.2 [118] to collect unaligned reads. Non-rRNA Ribo-seq reads and corresponding RNA-seq reads were then mapped to mouse mm10 reference genome using STAR v2.6.0c [112] and counted using featureCounts (Subread package v1.6.4; [119]). Reads were counted on the entire mRNA length for RNA-seq datasets, or only on CDS for Ribo-seq datasets to get footprints generated from actively translating ribosomes. Counts were normalized and translation efficiency was calculated using DESeq2 v1.24.0 [115] in R.

## Poly(A) tail length assay

Total RNA was extracted from P16 testicular extracts prepared as described above for ribosome profiling, using the Direct-zol RNA Miniprep kit (Zymo Research, R2050), including the in-column DNase I treatment. 500 ng of RNA were subjected to poly(U) tailing using poly(U) Polymerase (New England Biolabs, M0337S) in 1X NEBuffer 2 for 30 minutes at 37˚C. 160 ng of polyuridylated RNA were processed for cDNA synthesis using an RT adapter (GCG AGC ACA GAA TTA ATA CGA CTC ACT ATA GGA AAA AAA AAA AAT T) and the Super-Script III first-strand synthesis system (Invitrogen, 18080051), according to manufacturer's instructions. The obtained cDNA samples were PCR amplified using a forward primer mapping to the 3'UTR of the gene of interest and a reverse primer annealing to the RT adapter. Primers were used at a final concentration of 800 nM each and have the following sequences: Sycp1-3'UTR-For: GAG AGC CAA ACT TTA CCA AGG A; Setx-3'UTR-For: ACC TGT CTT ACT TTG CTT CTC TC; Adapter-Rev: GCG AGC ACA GAA TTA ATA CGA CT. PCR amplifications were performed using GoTaq DNA Polymerase (Promega, M3001), according to manufacturer's instructions; samples were visualized on 1.5% agarose gels.

## Retrotransposition assay

L1 retrotransposition assay was performed as previously described [33] with minor modifications. Briefly, HeLa cells were seeded in 6-well plates in supplemented DMEM culture medium. Seeding densities were optimized to obtain quantifiable G418-resistant colonies in every condition. For assays employing native mouse wild type L1 (pTN201) and for negative controls (synthetic mouse mutant L1 (pCEPsmL1mut) and mock treatment), 2 x $10^4$ cells were plated in each well. For assays employing synthetic mouse wild type L1 (pCEPsmL1), 800 cells were plated per well. For the control reporter plasmid (pPAGFP-C1), 2 x $10^3$ cells were plated per well.

The next day cells were transfected with 500 ng of L1 plasmids plus equimolar amounts of either pEGFP or pPRKRA constructs, diluted in 77 μl of Opti-MEM (Gibco, 31985062), using 2 μl of Fugene HD transfection reagent (Promega). The control reporter plasmid was transfected in parallel in equimolar amounts with pEGFP or pPRKRA. Transfection was blocked after 7 hours by replacing the medium with fresh DMEM.

48 hours post-transfection, live cells were imaged at an Eclipse Ti2 inverted microscope (Nikon) to determine the transfection efficiency for every experimental condition. Both brightfield and GFP images were acquired for the same cells, allowing a direct estimation of the number of EGFP[+] over the total number of cells. 72 hours post-transfection, G418 selection (400 μg/ml; Gibco, 10131035) was started. Medium was replaced every day until 14 days post-transfection. G418-resistant colonies were then fixed with 4% PFA in PBS for 1 hour and 30 minutes at room temperature and stained with 0.1% crystal violet (Sigma, V5265) in water.

Images of whole wells containing stained colonies were acquired on a Zeiss AxioZoom.V16 macroscope with Zen software (Zeiss). Using the Integrated Morphometry Analysis module of MetaMorph software (Molecular Devices), the images were segmented to separate colonies from isolated cells and irregular non-colony clusters. Colonies were identified on the basis of size, shape and uniformity of staining (minimum diameter 194.5 μm, circularity $\geq$ 0.45, gray level texture difference moment $\leq$ 2.75 [120]), and then counted. Over- and under-segmented colony counts were corrected by manual proofreading. Colony counts were obtained from three technical replicates per experimental condition. Mean colony counts were calculated and further adjusted for transfection efficiency to obtain the adjusted retrotransposition mean. The adjusted retrotransposition mean obtained for native mouse L1 (pTN201) was then divided by the value obtained for the control reporter plasmid (pPAGFP-C1) to calculate the corrected retrotransposition mean. The corrected retrotransposition mean was further converted to corrected retrotransposition efficiency to estimate the retrotransposition efficiency of pTN201 in the presence of pPRKRA, in comparison to pEGFP control (set as 100%).

## Supporting information

**S1 Fig. L1 ORF1p aggregates can be detected in the cytoplasm of spermatocytes from BALB/c wild-type mice. (A)** Immunofluorescence staining of L1 ORF1p (green) on BALB/c wild-type spermatocytes showing ORF1p accumulation in small cytoplasmic granules similar to early LBs. **(B–B')** Electron micrographs of a BALB/c wild-type spermatocyte harboring a small LB; boxed area in B is magnified in B'.
(TIF)

**S2 Fig. L1 ORF1p can be efficiently pulled-down from either unfractionated testicular extracts or pools of sucrose fractions where L1 RNP complexes sediment. (A)** Silver staining and **(B)** western blot analysis of anti-ORF1p immunoprecipitation samples (IP) from unfractionated and sucrose gradient fractionated (+ EDTA; pool sucrose fractions 5–8) $Mael^{-/-}$ testis extracts. Immunoprecipitations with carrier beads only (BO) were performed in parallel for background estimation; samples obtained from $Mael^{+/-}$ mice are shown as negative controls. Super: supernatant.
(TIF)

**S3 Fig. Genomic repeats and mRNAs detected across anti-L1 ORF1p co-immunoprecipitation samples. (A)** Extensive heatmap of repeat families detected across anti-L1 ORF1p co-immunoprecipitation samples: $Mael^{-/-}$ testis extracts (TOTAL), pooled sucrose fractions 5–8 (INPUT), carrier beads only (BO), ORF1p immunoprecipitated samples (IP). Average expression level of ancestral L1Lx family in TOTAL samples was chosen as the threshold to exclude any repeat with a lower expression from the reported heatmap. For simplicity, low complexity regions, rRNA, tRNA, satellites and simple repeats were also excluded. **(B)** Gene Ontology analysis (Biological Process) of the set of mRNAs (n = 1347) that were found strongly enriched in IP samples. Barplot shows the 13 most representative enriched GO terms (y axis, Term), with the corresponding number of genes identified per term (x axis, Count). Bars are labeled based on the False Discovery Rate (FDR).
(TIF)

**S4 Fig. Negative and positive controls for HCR RNA-FISH. (A)** Multiplexed HCR RNA-FISH of MMERVK10c-int and Setx with L1 RNA on $Mael^{-/-}$ testis sections minus (-) and plus (+) RNase A treatment. The RNase treatment is shown as a negative control for effective probe-RNA recognition. Scale bars: 40 μm. **(B)** HCR RNA-FISH of U6 small nuclear RNA on $Mael^{+/-}$ and $Mael^{-/-}$ testes. U6 small nuclear RNA was chosen as a positive control for its high

abundance and strict nuclear localization. As expected, U6 shows strong nuclear signals, independent from the *Mael* mutation. Scale bars: *Mael*[+/-] section: 20 μm; *Mael*[-/-] section: 10 μm. (TIF)

**S5 Fig. Optimization of RNase treatment of testis extracts before co-immunoprecipitation.** **(A)** Non-denaturing RNA agarose gel showing multiple RNase treatment conditions (lanes 3–6, see box for details) or untreated BL/6 testis total RNA (lane 2) as a control; the treatment used in lane 5 was chosen for co-immunoprecipitation experiments. Approximately 2 μg of RNA were loaded on each lane. **(B)** Bioanalyzer traces of RNA samples corresponding to lanes 2 and 5 in panel A. RIN: RNA Integrity Number; scale from 1 (fully degraded) to 10 (intact). **(C)** Silver staining of anti-ORF1p co-immunoprecipitation samples (IP) from *Mael*[-/-] testicular extracts minus (-) and plus (+) the RNase treatment used for lane 5 in panel A. Immunoprecipitation with an isotype IgG served as a negative control; note that the RNase treatment increases immunoprecipitation of ORF1p. **(D)** Heatmap showing the recovery of ribosomal proteins in anti-ORF1p co-immunoprecipitation samples from *Mael*[-/-] testes, minus (-) and plus (+) the RNase treatment used for lane 5 in panel A. Color indicates protein levels; E2.n indicates Experiment 2.replicate n. The relative abundance of each protein in each eluate was obtained by mass spectrometry analysis and further adjusted by dividing the PSM counts by molecular weights (MW); for a sample-to-sample comparison, the obtained PSM / MW ratios were normalized to the ratio obtained for the bait (LORF1). (TIF)

**S6 Fig. Localization of CNOT1 and DCP1A proteins in mouse *Mael*[+/-] and *Mael*[-/-] testes.** **(A)** Double immunofluorescence staining of CNOT1 (green) and L1 ORF1p (magenta) in *Mael*[+/-] and *Mael*[-/-] testes. In *Mael*[+/-], CNOT1 shows aggregation in characteristic round cytoplasmic structures in spermatocytes (top panels, yellow arrows) and in more restricted cytoplasmic areas in round spermatids (top panels, red arrows). Similar structures are also observed in spermatocytes of *Mael*[-/-] mice (bottom panels, yellow arrows), where L1 ORF1p is not detected, in addition to some CNOT1 aggregation in LBs (bottom panels, white arrowheads). Scale bars: 15 μm. **(B)** Double immunofluorescence staining of DCP1A (green) and L1 ORF1p (magenta) in *Mael*[+/-] and *Mael*[-/-] testis sections. DCP1A shows a cytoplasmic granular pattern that is detected in all spermatogenic cells of *Mael*[+/-] control testes (top panels); signal intensities and sizes of granules are variable. In *Mael*[-/-] tubules (bottom panels), DCP1A is mostly confined to prominent granules with strong signals in peripheral spermatogonia; weak diffuse DCP1A signal is observed in LBs. Scale bars: 15 μm. (TIF)

**S7 Fig. PRKRA association with L1 ORF1p is recapitulated in F9 mouse carcinoma cells using an exogenous construct.** **(A)** Immunofluorescence staining of PRKRA (green) in *Mael*[-/-] testis. PRKRA shows diffuse cytoplasmic signals in both germ (spermatogonia and spermatocytes) and somatic (Sertoli) cells; PRKRA aggregation in LBs-resembling structures in spermatocytes is also detected (yellow arrows). Scale bar: 15 μm. **(B)** Schematic representation of the pPRKRA plasmid driving the expression of N-terminal 3XFLAG mouse PRKRA followed by the self-cleaving P2A peptide and an EGFP reporter, and its control empty vector pEGFP. **(C)** Western blot analysis of exogenous PRKRA and endogenous ORF1p in F9 cells transfected with pEGFP or pPRKRA. Exogenous PRKRA was detected by both an anti-PRKRA and an anti-FLAG antibody with overlapping signals (red and green, respectively; left blot) at the expected MW (~ 37 kDa); no detectable perturbation of endogenous ORF1p was observed upon PRKRA overexpression (right blot). cyt: cytoplasmic extract; nuc: nuclear extract. **(D)** Double immunofluorescence staining of 3XFLAG-PRKRA (anti-FLAG antibody)

and L1 ORF1p in F9 cells transfected with pEGFP or pPRKRA. Boxed areas in pEGFP panels are magnified in corresponding insets and identify a cell with a relatively high EGFP expression; besides a major nuclear localization due to an added nuclear localization signal (NLS), EGFP shows a diffuse cytoplasmic distribution independent of L1 ORF1p granules. Boxed areas in pPRKRA panels are magnified in corresponding insets and identify a cell with a moderate expression level of pPRKRA plasmid; exogenous 3XFLAG-PRKRA shows a cytoplasmic distribution that partially overlaps with L1 ORF1p granules (yellow arrow). Scale bars: 20 μm. **(E)** Western blot analysis of anti-ORF1p co-immunoprecipitation samples obtained from F9 cells transfected with pEGFP or pPRKRA; 3XFLAG-PRKRA co-precipitates with endogenous ORF1p. Immunoprecipitations with an isotype IgG served as negative controls. **(F)** Additional positive and negative controls for L1 retrotransposition assay. Synthetic mouse wild type (pCEPsmL1) and corresponding retrotransposition-defective control (pCEPsmL1mut) L1 elements are transfected in HeLa cells in combination with pEGFP or pPRKRA. High retrotransposition efficiency is observed using the positive control pCEPsmL1 plasmid either with pEGFP or pPRKRA; no neomycin-resistant colonies (neo$^R$) are generated from cells transfected with pCEPsmL1mut and pPRKRA. These data confirm that L1 retrotransposition proceeds unperturbed in the presence of PRKRA and that the obtained colonies are derived from authentic retrotransposition.
(TIF)

**S1 Table. mRNAs bound by L1 ORF1p in *Mael*$^{-/-}$ testes (n = 2081) with their corresponding enrichment in IP samples (log2FoldChange and p-adjusted values); related to Fig 3B, 3C and 3D.**
(XLSX)

**S2 Table. Gene Ontology analysis (Biological Process) of the set of mRNAs strongly enriched in anti-ORF1p IP samples (n = 1347); related to Figs 3B and S3B.**
(XLSX)

**S3 Table. Mass spectrometry analysis of L1 ORF1p interactors: raw data and proteins retained after background subtraction from two independent experiments (E1, E2); related to Fig 4.**
(XLSX)

**S4 Table. High confidence L1 ORF1p interactors (n = 80); related to Fig 4.**
(XLSX)

**S5 Table. Comparison of L1 ORF1p interactors identified in this study (mouse proteins) with previously published human datasets; related to Fig 4.**
(XLSX)

## Acknowledgments

We thank Allison Pinder, Dr. Fred Tan, Dr. Xiaobin Zheng and Dr. Sarah Wheelan for assistance with sequencing and data analyses; Dr. Eugenia Dikovsky and the Animal Facility staff for assistance with animal care; Dr. Kamena Kostova for advice on ribosome profiling; Rejeanne Juste for assistance with genotyping and tissue sectioning; Dr. Svetlana Deryusheva for reagents and assistance with RNA FISH optimization; Dr. Mahmud Siddiqi for assistance with microscopy and automated colony counting in retrotransposition assays; Michael Sepanski for assistance with EM imaging. We thank Dr. Joseph Tran, Dr. Zhao "ZZ" Zhang and past and present members of the Bortvin lab for discussions and suggestions. We thank Dr. John Goodier and Dr. Svetlana Deryusheva for their feedback on the manuscript.

## Author Contributions

**Conceptualization:** Chiara De Luca, Alex Bortvin.

**Data curation:** Chiara De Luca, Anuj Gupta.

**Formal analysis:** Chiara De Luca, Alex Bortvin.

**Funding acquisition:** Alex Bortvin.

**Investigation:** Chiara De Luca, Alex Bortvin.

**Project administration:** Alex Bortvin.

**Resources:** Alex Bortvin.

**Software:** Anuj Gupta.

**Supervision:** Alex Bortvin.

**Validation:** Chiara De Luca, Alex Bortvin.

**Visualization:** Chiara De Luca, Alex Bortvin.

**Writing – original draft:** Chiara De Luca, Alex Bortvin.

**Writing – review & editing:** Chiara De Luca, Alex Bortvin.

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
