## [Decision Letter · Decision Letter 0]

28 Mar 2023

Dear Dr Bortvin,

Thank you very much for submitting your Research Article entitled 'Ribonucleoprotein condensation driven by retrotransposon LINE-1 sustains RNA integrity and translation in mouse spermatocytes' to PLOS Genetics.

The manuscript was fully evaluated at the editorial level and by three independent peer reviewers. The reviewers appreciated the attention to an important topic and they all agree that the study is of high quality and brings new important insights. The reviews identified some issues that we ask you address in a revised manuscript. They also make a number of valuable comments on specific sections that we would like you to consider. 

We therefore ask you to modify the manuscript according to the review recommendations. Your revisions should address the specific points made by each reviewer.

Yours sincerely,

Cédric Feschotte

Academic Editor

PLOS Genetics

Gregory Barsh

Editor-in-Chief

PLOS Genetics

Reviewer's Responses to Questions

**Comments to the Authors:**

Reviewer #1: In this manuscript by De Luca et. al. the authors investigate the composition of LINE-1 bodies in Mael -/- spermatocytes by performing experiments aimed at identifying ORF1p bound RNAs in LB enriched fractions. They also perform experiments to identify novel ORF1p interactors in LBs. The authors find that LBs contain fully assembled ribosomes, yet the interaction of L1 RNA and ORF1p within LBs does not require the presence of ribosomes. They also show that ORF1p associates with PRKRA, which they demonstrate to be a novel activator of L1 retrotransposition. Overall, the manuscript is well written (with a very thoughtful discussion section, in particular) and the data is very interesting. I have a few concerns about controls used in experiments, and I am curious about a few details regarding the LBs, as detailed below.

1) RNA-seq was performed on ORF1p immunoprecipitated RNAs in Figure 3. In 3A, the authors compare the abundance of RNAs in the ORF1p IP to the RNAs in the INPUT, the TOTAL extracts and the Beads only (BO). I am worried that an important specificity control is missing, namely an IgG immunoprecipitation control. Typically, in RNA-imunoprecipitation experiments, IgG immunoprecipitation will be performed to rule out non-specific interactions. Have the authors performed this control? It seems like it is mentioned in the methods, and in supplemental tables, but I did not see it in the actual experimental analyses. The authors do confirm co-localization of two of their top hits with the ORF1p in LBs, but I am more concerned with the conclusions made from the data analyses in A-D. If the pool of immunoprecipitated RNAs is too large due to insufficient controls, then important insights could be missed, including in later figures when the length of ORF1p associated RNAs is examined.

2) Also, in Fig. 3A, could the authors explain why L1 RNA is more abundant in the ORF1p IP than in the INPUT? Are the input samples actually a percent of the input used in the IP experiments, or are they the remaining supernatant following IP from those fractions?

3) On page 11, the authors make the statement that “L1 overexpression and LBs formation disrupt PBs formation.” Do the authors have any biochemical or quantitative data to support this strong statement? If not, they may want to consider rewording.

4) On page 12, the authors claim their findings “suggest that the CCR4-NOT complex does not deadenylate LB-localized mRNAs, based on their data that two ORF1p associated mRNAs, Sycp1 and Setx, exhibit no changes in polyA tail length in Mael -/- testes as compared to controls. To support their claim, the authors would need to show that CCR4-NOT associates with these RNAs in both Mael -/- and control testes.

5) The EM experiments in Fig 1 are beautiful and it is interesting that the LBs from Mael -/- spermatocytes are often double membraned structures. Do the LBs in WT spermatocytes also have double membranes? Have the authors investigated whether these are some sort of ER-derived organelles? These could be interesting points to touch on in the discussion.

Reviewer #2: Long Interspersed Element 1 (L1) is an active retrotransposon comprising ~18% of the mouse genome. As L1s are selfish genetic elements, they must retrotranspose in cells of the germ line or early embryonic cells, where new L1 insertions have the potential to be inherited by subsequent generations. To curtail deleterious L1 activity, a piRNA based adaptive defence mechanism has evolved in the mouse male germ line. Numerous factors are required for piRNA biogenesis, including the RNA-binding protein Maelstrom (Mael). Mael-/- germ cells overexpress L1 ORF1p, which localizes to large cytoplasmic aggregates here termed L1 bodies, or LBs.

In this study, De Luca et al. investigated the proteins and RNAs that interact with L1 ORF1p in LBs. They find that LBs are enriched in L1 RNA, ORF1p, and ribosomes, that L1 RNA and ORF1p interact in both ribosome-associated and ribosome-independent complexes. Analysis of RNAs associated with ORF1p revealed active L1 subfamily RNAs and a variety of cellular mRNAs. IP-mass spec analysis of ORF1p interacting proteins from Mael-/- testicular extracts uncovered both known and novel interactors, including members of the CCR4-NOT complex, and strengthened the conclusion that LBs share some characteristics with but are distinct from both stress granules and P-bodies. Notably, aggregation in LBs did not appear to impact mRNA stability, poly-A tail length, or translation. Intriguingly, the RNA-independent ORF1p interactor PRKRA, which can both enhance and inhibit translation, was found to increase ORF1p levels in an ectopic overexpression system in HeLa cells.

This work is important as it characterises the L1 ORF1p protein and RNA interactome in a cell type that is relevant to the generation of heritable L1 insertions and the impact of L1 activity on genome evolution, using Mael deficiency as a model of L1 overexpression. The experiments presented are of high quality and in many ways confirm the results of previous studies, while revealing added complexity in the interaction between L1 and host factors within cytoplasmic aggregates. The most interesting implication, based on PRKAR’s ability to increase L1 ORF1p levels and the finding that LBs are not associated with mRNA degradation, is that association of L1 RNPs with cellular factors in cytoplasmic inclusions, at least in germ cells, may, in part, represent an adaptive mechanism to evade host cell defences, rather than simply a sequestration of the L1 machinery by the host cell to limit retrotransposition.

Points for consideration:

1. This work entails an extensive characterization of LBs, but these structures are specific to germ cells where L1 is overexpressed due to, in this case, Mael deficiency. It would be quite interesting to examine the L1 ORF1p interactome in wild-type germ cells, which do contain ORF1p aggregates (as shown in Figure S1), in comparison to Mael-/- cells. However these experiments would be incredibly challenging given the low levels of L1 ORF1p expression in wild-type germ cells. Perhaps the authors could speculate on extent to which the molecular interactions occurring in LBs in the presence of extreme L1 overexpression might also take place in ORF1p aggregates under normal L1 expression levels. In other words, how relevant are these results to L1 activity in wild-type testis?

2. In figure 3A, the inclusion active L1 GF subfamily elements and B1 SINEs in the heat map of repeat RNAs found in germ cell ORF1p complexes would be interesting, as these elements are also known to be active in mice.

3. Can the authors speculate on the strong enrichment for MMERVK10C-int RNA in ORF1p complexes? This result is unexpected and quite interesting, yet it is not clear why this RNA would associate so strongly with ORF1p. For comparison, it would be helpful to include another active LTR retrotransposon (an IAP family for example) from the full heatmap in Figure S3, to highlight the unique enrichment for MMERVK10C-int.

4. Line 71: LTR retrotransposons, not L1s, account for 10-15% of novel germline mutations in mice.

5. Line 344: the primary reference for the synthetic mouse L1 constructs is Han and Boeke 2004 (reference 109).

6. Line 358: a new study characterising the L1 interactome could be added to the listed references for proteomic analysis of L1 RNP interactors (PMID: 36639706).

Reviewer #3: L1 ribonucleoprotein (RNP) complex is an obligatory intermediate for its replication and propagation. Many important insights about its subcellular localization and molecular composition have been gleaned from overexpressing L1 in cultured cells and most recently from prostate cancer cells by using RNA immunoprecipitation (ref 63). In this context, profiling the complexity of RNAs and their fate in the so called “L1 bodies” in mouse germ cells represents a critical effort in our understanding of not only host mechanisms of L1 regulation but also roles of L1 RNP in host gene expression. In this manuscript, De Luca et al utilized an assortment of molecular, cellular and ultracellular techniques to dissect the composition of L1 bodies and provided an initial assessment of the potential function of such complex. Overall, the experiments were well executed, and data were clearly presented, only with a few blemishes concerning some aspects of overinterpretation of the role of L1 bodies and PRKRA experiments.

Major comments:

1. The authors did an outstanding job in demonstrating L1 body morphology in Mael-/- spermatocytes with EM, the presence of selected ribosomal proteins with immunofluorescence, differential association of L1 ORF1p and polyribosome with sedimentation, the identification of TE RNA and mRNA species with RNA immunoprecipitation and sequencing, the identification of (new) protein partners interacting with L1 RNP with mass spectrometry, and the characterization of RNA integrity and translation efficiency. These experiments were carefully designed, executed, and analyzed. The comprehensive identification of RNA and protein components in L1 bodies in mouse spermatocytes is novel and lays foundation for future studies.

2. However, this reviewer feels that the role of L1 bodies may have been overinterpreted. As stated in the title and at the end of the abstract, the authors conclude that L1 body sustains RNA integrity and translation of endogenous RNAs in mouse spermatocytes. However, the data about the abundance of endogenous RNAs in L1 body and these RNAs’ stability are correlational. Indeed, the authors also noted the caveat that these RNAs might only represent a fraction of the total amount for each RNA species. Importantly, the Mael-/- cells cannot progress through meiosis, and are not “normal” environment for L1 propagation.

3. The authors further cauterized potential role of PRKRA by overexpressing it in HeLa cells in the presence of an L1spa reporter plasmid. They showed a slight decrease in L1spa RNA but an increase in ORF1p and L1spa retrotransposition. The authors did not check L1 body formation and colocalization of PRKRA with ORF1p in these cells. So it remains premature to conclude that ORF1p condensate promotes L1 retrotransposition.

4. In the conclusion section (page 17,18), the authors further suggest that L1 body represents an active mechanism to counteract cellular defenses as sites of active translation. This hypothesis is very interesting and quite profound but requires additional experimental support. It should be noted that these L1 bodies form in Mael-/- germ cells, which are an evolutionary dead end as they lose the capacity of differentiating into mature germ cells. So the system, while presenting abundant L1 bodies, has its limitations.

Minor comments:

1. Figure 1. Some cellular landmarks should be labelled in the EM images to increase readability.

2. Figure 3A. Color key says 14, 15, and 16.

3. Figure 3E. Spermatocytes and spermatids should be marked.

4. Ref 27 is misplaced.

**Have all data underlying the figures and results presented in the manuscript been provided?**

Reviewer #1: Yes

Reviewer #2: Yes

Reviewer #3: Yes

PLOS authors have the option to publish the peer review history of their article (what does this mean?). If published, this will include your full peer review and any attached files.

Reviewer #1: No

Reviewer #2: No

Reviewer #3: No

---

## [Editor Report · Decision Letter 1]

23 May 2023

Dear Dr Bortvin,

We are pleased to inform you that your manuscript entitled "Retrotransposon LINE-1 Bodies in the Cytoplasm of piRNA-Deficient Mouse Spermatocytes: Ribonucleoproteins Overcoming the Integrated Stress Response" has been editorially accepted for publication in PLOS Genetics. Congratulations!

Yours sincerely,

Cédric Feschotte

Academic Editor

PLOS Genetics

Gregory Barsh

Editor-in-Chief

PLOS Genetics

Comments from the reviewers (if applicable):

**Data Deposition**

http://datadryad.org/submit?journalID=pgenetics&manu=PGENETICS-D-23-00097R1

**Press Queries**

---

## [Editor Report · Acceptance letter]

7 Jun 2023

PGENETICS-D-23-00097R1 

Retrotransposon LINE-1 Bodies in the Cytoplasm of piRNA-Deficient Mouse Spermatocytes: Ribonucleoproteins Overcoming the Integrated Stress Response 

Dear Dr Bortvin, 

We are pleased to inform you that your manuscript entitled "Retrotransposon LINE-1 Bodies in the Cytoplasm of piRNA-Deficient Mouse Spermatocytes: Ribonucleoproteins Overcoming the Integrated Stress Response" has been formally accepted for publication in PLOS Genetics! Your manuscript is now with our production department and you will be notified of the publication date in due course.

With kind regards,

Anita Estes

PLOS Genetics

On behalf of:
